# The Ca$^{2+}$ transient as a feedback sensor controlling cardiomyocyte ionic conductances in mouse populations

Colin M Rees[1,2†], Jun-Hai Yang[3,4†], Marc Santolini[1,2], Aldons J Lusis[3,4,5,6], James N Weiss[3,4], Alain Karma[1,2*]

[1]Physics Department, Northeastern University, Boston, United states; [2]Center for Interdisciplinary Research on Complex Systems, Northeastern University, Boston, United States; [3]Department of Medicine (Cardiology), Cardiovascular Research Laboratory, David Geffen School of Medicine, University of California, Los Angeles, United states; [4]Department of Physiology, David Geffen School of Medicine, University of California, Los Angeles, United States; [5]Department of Microbiology, David Geffen School of Medicine, University of California, Los Angeles, United States; [6]Department of Human Genetics, David Geffen School of Medicine, University of California, Los Angeles, United States

*For correspondence:
a.karma@northeastern.edu

†These authors contributed equally to this work

Competing interests: The authors declare that no competing interests exist.

**Abstract** Conductances of ion channels and transporters controlling cardiac excitation may vary in a population of subjects with different cardiac gene expression patterns. However, the amount of variability and its origin are not quantitatively known. We propose a new conceptual approach to predict this variability that consists of finding combinations of conductances generating a normal intracellular Ca$^{2+}$ transient without any constraint on the action potential. Furthermore, we validate experimentally its predictions using the Hybrid Mouse Diversity Panel, a model system of genetically diverse mouse strains that allows us to quantify inter-subject versus intra-subject variability. The method predicts that conductances of inward Ca$^{2+}$ and outward K$^+$ currents compensate each other to generate a normal Ca$^{2+}$ transient in good quantitative agreement with current measurements in ventricular myocytes from hearts of different isogenic strains. Our results suggest that a feedback mechanism sensing the aggregate Ca$^{2+}$ transient of the heart suffices to regulate ionic conductances.

DOI: https://doi.org/10.7554/eLife.36717.001

## Introduction

Following the landmark publication of the Hodgkin-Huxley model of nerve-cell action potential over six decades ago (*Hodgkin and Huxley, 1952*), electrophysiological models of increasing complexity have been developed to describe the cardiac action potential (AP) and its interaction with the intracellular calcium (Ca$^{2+}$) signal (*Noble, 2011*; *Silva and Rudy, 2010*), which links electrical signaling to mechanical contraction in cardiomycoytes (*Bers, 2001*). As illustrated in *Figure 1* for a mouse ventricular mycoyte (*Bondarenko et al., 2004*), those models typically involve a large set of interacting cellular components that includes various voltage-gated membrane ion channels and transporters, the Na$^+$/Ca$^{2+}$ exchanger, and Ca$^{2+}$ handling proteins such as the ryanodine receptor (RyR) Ca$^{2+}$ release channels, which open in response to Ca$^{2+}$ entry into the cell via L-type Ca$^{2+}$ channels, and the sarcoplasmic reticulum (SR) Ca$^{2+}$ ATPase (SERCA), which uptakes Ca$^{2+}$ back into the SR. Ca$^{2+}$ release and uptake from the SR causes a transient rise in cytosolic Ca$^{2+}$ concentration, the calcium transient (CaT), which activates myocyte contraction. Those cellular-scale models have been traditionally constructed by piecing together separate mathematical models describing molecular-scale

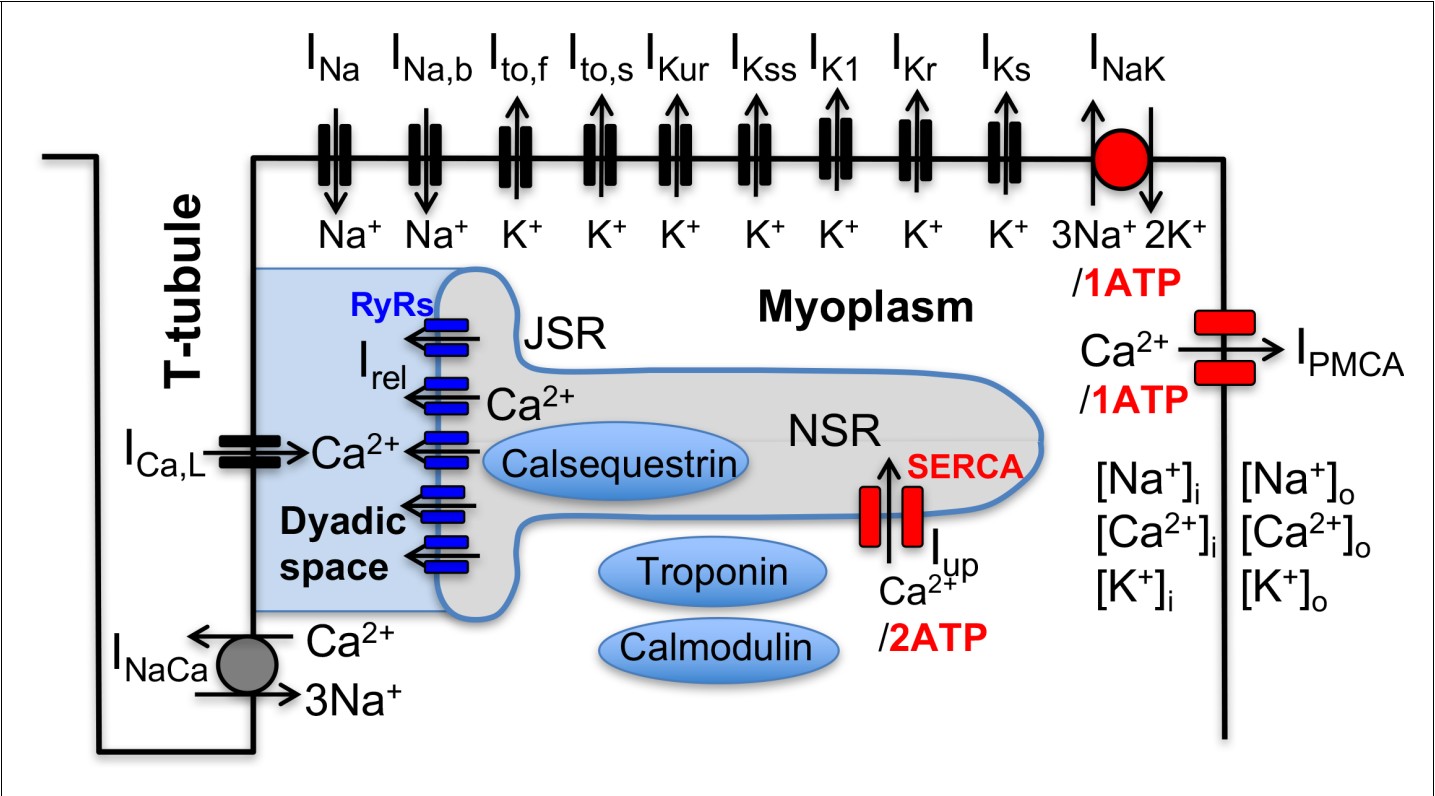

**Figure 1.** Schematic representation of the sarcolemmal currents and intracellular $Ca^{2+}$ cycling proteins of the mouse ventricular myocyte model.
DOI: https://doi.org/10.7554/eLife.36717.002

components in the same species (guinea pig (*Luo and Rudy, 1991*), mouse (*Bondarenko et al., 2004*), rabbit (*Shannon et al., 2004*; *Mahajan et al., 2008*), dog (*Fox et al., 2002*), etc.), and by sometimes mixing models in different species. Experimental measurements of voltage-current relationships and other properties used to develop those models are typically averaged over several cells from one or a few hearts. While such prototypical 'population-averaged' (and even 'species-averaged') models have proven useful to investigate basic mechanisms of cardiac arrhythmias on cellular to tissue scales (*Krogh-Madsen and Christini, 2012*; *Karma, 2013*; *Qu et al., 2014*), they fall short of predicting how different individuals in a genetically diverse population respond to perturbations (such as physiological stressors, ion channel mutations, drug or gene therapies, etc.) affecting one or several cellular components.

In a neuroscience context, the limitation of population-averaged models has been highlighted by pioneering theoretical and experimental studies by Abbot, Marder, and co-workers demonstrating that ion channel conductances can exhibit a high degree of activity-dependent plasticity as well as variability between individuals (*LeMasson et al., 1993*; *Siegel et al., 1994*; *Liu et al., 1998*; *Golowasch et al., 1999*; *Prinz et al., 2004*; *Schulz et al., 2006*; *Marder and Goaillard, 2006*; *Grashow et al., 2009*; *Marder, 2011*; *O'Leary et al., 2013*). Theoretical studies along this line first originated from an attempt to explain why neurons can maintain fixed electrical activity patterns despite a high rate of ion channel turnover. This question was addressed by treating ion channel conductances as dynamical variables in models of neuronal activity and by using the intracellular $Ca^{2+}$ concentration as an activity-dependent feedback mechanism regulating their values (*LeMasson et al., 1993*; *Siegel et al., 1994*; *Liu et al., 1998*), a mechanism supported by experiments (*Golowasch et al., 1999*). Those model neurons displayed remarkable properties such as the ability to modify their conductances to maintain a given behavior when perturbed or to develop different properties in response to different patterns of presynaptic activity. Subsequently, a different type of computational study in which model parameters (conductances and synaptic strengths of a circuit model of the crustacean somato-gastric ganglion) were varied randomly, demonstrated that a

similar bursting activity could be obtained with multiple parameter sets, that is multiple 'good enough solutions' (GES) (*Prinz et al., 2004*). This prediction agreed qualitatively with an experimental study demonstrating that neurons of the same circuits obtained from different crabs could have markedly different ion channel densities, corresponding to different gene expression, but yet different circuits could generate similar bursting activities (*Schulz et al., 2006*).

A later experimental study further showed that different circuits, while operating similarly under controlled conditions, could respond differently to perturbations such as serotonin addition, which increases the bursting frequency at a population level, but lowers it in some individuals (*Grashow et al., 2009*). Those findings shed light on why pharmacological treatments may work in some individuals but not others. In addition, they suggest that the existence of different good enough solutions provides an evolutionary advantage for the survival of a genetically diverse population by allowing different individuals to better adapt to different environmental challenges so as to survive and restore the population.

Those studies and related ones in the broader context of systems biology (*Daniels et al., 2008*; *Gutenkunst et al., 2007*; *Transtrum et al., 2015*) and cardiac electrophysiology (*Sarkar and Sobie, 2009*; *Sarkar and Sobie, 2010*; *Weiss et al., 2012*; *Sarkar et al., 2012*; *Britton et al., 2013*; *Groenendaal et al., 2015*; *Muszkiewicz et al., 2016*; *Krogh-Madsen et al., 2016*) have produced a paradigm shift from population-averaged models, with unique fine-tuned parameter sets, to populations of models characterized by multiple parameter sets. In this new paradigm, each set representing a different individual in a population can produce a similar behavior under controlled conditions, but a starkly different response to perturbations for some individuals. This paradigm shift, however, creates new theoretical and experimental challenges.

On the theoretical side, an open question is how to search for GES representing different individuals in a population. The results of this search, which has been carried out using various methods (e.g. random search (*Prinz et al., 2004*), multivariate regression analysis (*Sarkar and Sobie, 2009*; *Sarkar and Sobie, 2010*), or genetic algorithms (*Groenendaal et al., 2015*), depend critically on what outputs are selected to constrain model parameters. To date, parameter searches in a cardiac context have been 'AP centric', focusing primarily on features of the membrane voltage ($V_m$) signal. Sarkar and Sobie showed that very different combinations of ion conductances can produce almost identical cardiac AP waveforms, albeit different CaT amplitudes, and that adding additional constraints on the $V_m$ and $Ca^{2+}$ signals can further constrain model parameters (*Sarkar and Sobie, 2009*; *Sarkar and Sobie, 2010*). Groenendaal et al., 2015 constrained model parameters using $V_m$ traces with variable AP waveforms recorded from guinea pig cardiomyocytes under randomly timed electrical stimuli, as opposed to a unique AP waveform recorded during periodic pacing. This search yielded parameter sets that are potentially better suited to describe more complex aperiodic forms of $V_m$ dynamics relevant for arrhythmias. *Britton et al., 2013* observed experimentally a significant variability in AP waveform in rabbit Purkinje fibers and searched for model parameter combinations that reproduce this waveform variability. They then used those parameter sets to predict different effects of pharmacological blockade of cardiac HERG ($I_{Kr}$ current) potassium channel in different subjects (*Britton et al., 2013*).

All those GES searches have relied for the most part on using measured AP features (*Sarkar and Sobie, 2009*; *Sarkar and Sobie, 2010*; *Sarkar et al., 2012*; *Britton et al., 2013*; *Groenendaal et al., 2015*; *Muszkiewicz et al., 2016*; *Krogh-Madsen et al., 2016*) to constrain model parameters, even though some recent studies have also considered the additional effect of constraining the CaT (*Passini et al., 2016*; *Mayourian et al., 2017*). However, given that there is no known voltage-sensing mechanisms regulating ion channel expression, it remains unclear if natural biological variability can be predicted based on AP features. Here, we adopt a different '$Ca^{2+}$ centric' view (*Weiss et al., 2012*), which postulates as in a neuroscience context (*Golowasch et al., 1999*; *LeMasson et al., 1993*; *Siegel et al., 1994*; *Liu et al., 1998*; *O'Leary et al., 2013*) that model parameters are predominantly constrained by feedback sensing of $Ca^{2+}$, and potentially other ions (e.g. $Na^+$) affecting ion channel regulation. Our hypothesis is that the CaT is critical for generating blood pressure, which is sensed by the carotid baroreceptors and feeds back through the autonomic nervous system to regulate the CaT via controlling levels of Ca-cyling proteins and the AP in a way that preserves blood pressure. This provides a very straightforward physiological mechanism that we show not only constrains the CaT to a physiological waveform, but, as an added and novel bonus, also constrains AP features through the ratio of inward Ca currents and outward K currents. Under

this hypothesis, multiple parameter combinations producing a normal CaT could potentially represent different GES in a genetically diverse population. In addition, unlike voltage, intracellular concentrations of $Ca^{2+}$ and $Na^+$ ions ($[Ca]_i$ and $[Na]_i$, respectively) have a known interactive role in transcriptional regulation of cardiac ion channel proteins and their function (*Rosati and McKinnon, 2004*). For example, the $Ca^{2+}$/calcineurin/NFAT pathway regulates L-type $Ca^{2+}$ channel (LCC) expression (*Qi et al., 2008*) and $Na^+$ modulates cAMP-dependent regulation of ion channels in the heart (*Harvey et al., 1991*) including phosphorylation of LCCs via cAMP-dependent protein kinase (*Balke and Wier, 1992*). To test this hypothesis, we perform a GES search in which parameters of a mouse ventricular myocyte model are only constrained by CaT features and $[Na]_i$. This search yields GES with different conductances of the L-type $Ca^{2+}$ current ($I_{Ca,L}$) and $K^+$ currents ($I_{to,f}$ and $I_{Kur}$) and reveals that conductances are strongly correlated due to compensatory effects of those currents on the CaT.

On the experimental side, a major challenge is to test whether different GES produced by any given search method are representative of different individuals in a genetically diverse population. Performing this test generally requires distinguishing quantitatively the variability of conductances and electrophysiological phenotype observed in cells extracted from the same heart (intra-heart cell-to-cell variability) from the variability of the same quantities between different subjects (inter-subject variability). Making this distinction is made extremely difficult by the fact that AP features and conductances vary significantly between cells extracted from same region of the heart (*Banyasz et al., 2011*; *Groenendaal et al., 2015*) and that regional (e.g. ventricular base-to-apex and epicardium to endocardium) variations of ion channel expression are also present. The existence of large intra-heart cell-to-cell variability, and the practical limitation that only a finite number of cells can typically be extracted from a single heart for current measurements, raises the question of whether it is feasible to distinguish electrophysiological parameters between genetically distinct individuals.

To cope with this challenge, we use here the Hybrid Mouse Diversity Panel (HMDP) that is a collection of approximately 100 well-characterized inbred strains of mice that can be used to analyze the genetic and environmental factors underlying complex traits. Because inbred strains are isogenic and renewable, we are able to use multiple hearts from the same strain to obtain enough statistics to differentiate quantitatively between intra-heart and inter-subject variability in conductances of key currents ($I_{Ca,L}$, $I_{to,f}$ and $I_{Kur}$) affecting the AP and CaT of mouse ventricular myocytes from different strains. The results show that, despite large cell-to-cell variability, some strains have clearly distinguishable mean conductances (i.e. conductances averaged over all cells for the same strain). Mean conductances can vary by up to two-and-a-half fold between strains. The results further show that, remarkably, variations of mean conductances for individual strains follow the same correlation ($I_{Ca,L}$ current is large or small when the sum of $I_{to,f}$ and $I_{Kur}$ currents is large or small, respectively) predicted by our computational $Ca^{2+}$ centric GES search. The central hypothesis that parameters are constrained predominantly by features of the CaT (as a surrogate for arterial blood pressure) is further validated experimentally by showing that strains with very different conductances have similar contractile activity. It is worth emphasizing that the main novelty of the present study is the use of the HMDP to validate this hypothesis. The computational identification of GES itself uses a standard search algorithm, which consists of minimizing a cost function constructed from features of the CaT and the intracellular sodium concentration. In addition, we use tissue scale simulations to show that compensation remains effective at an organ scale despite large cell-to-cell variability within an individual heart. Finally, we use our findings to interpret the results of recent studies of cardiac hypertrophy and heart failure induced by a stressor in the HMDP (*Ghazalpour et al., 2012*; *Rau et al., 2015*; *Rau et al., 2017*; *Santolini et al., 2018*).

## Results

### Effects of individual conductances on the calcium transient

We first used a mouse ventricular myocyte model to investigate the effects of changing a single electrophysiological parameter on the CaT. This model (see Materials and methods) combines elements of previously published ventricular mycoyte models (*Shiferaw et al., 2003*; *Shannon et al., 2004*; *Bondarenko et al., 2004*; *Mahajan et al., 2008*). The CaT was characterized by its amplitude, defined as the difference $\Delta[Ca]_i$ between the diastolic and peak value of the cytosolic $Ca^{2+}$

concentration $[Ca]_i$, and the time-averaged value of $[Ca]_i$ over one pacing period, denoted by $\langle[Ca]_i\rangle$. The CaT amplitude $\Delta[Ca]_i$ is a major determinant of the contractile force while $\langle[Ca]_i\rangle$ provides an average measure of the cytosolic $Ca^{2+}$ concentration in the cell. Both quantities will be used as $Ca^{2+}$ sensors for our multi-parameter search of GES and examining individual parameter effects will be useful later to interpret the results of that search. We vary the conductances of sarcolemmal currents and transporters depicted in *Figure 2A* and the expression levels of $Ca^{2+}$ handling proteins that include the ryanodine receptor (RyR) $Ca^{2+}$ release channels and the sarcoplasmic reticulum (SR) $Ca^{2+}$ ATPase SERCA, which pumps $Ca^{2+}$ from the cytosol into the SR. For each parameter value, we pace the myocyte at a 4 Hz frequency for many beats until a steady-state is reached where the CaT profile used to calculate $\Delta[Ca]_i$ and $\langle[Ca]_i\rangle$ and the intracellular sodium concentration $[Na]_i$ no longer vary from beat to beat.

*Figure 2* shows the effects of individual parameter changes on the steady-state CaT amplitude (*Figure 2A*) and average $[Ca]_i$ (*Figure 2B*). Those six parameters were selected because they control the major currents influencing the CaT. Both quantities are plotted as a function of conductance fold change $G/G_{ref}$ where $G_{ref}$ is a reference value producing a normal CaT. Increasing the conductance of the inward L-type $Ca^{2+}$ current $I_{Ca,L}$ is seen to strongly increase both $\Delta[Ca]_i$ and $\langle[Ca]_i\rangle$ but has a

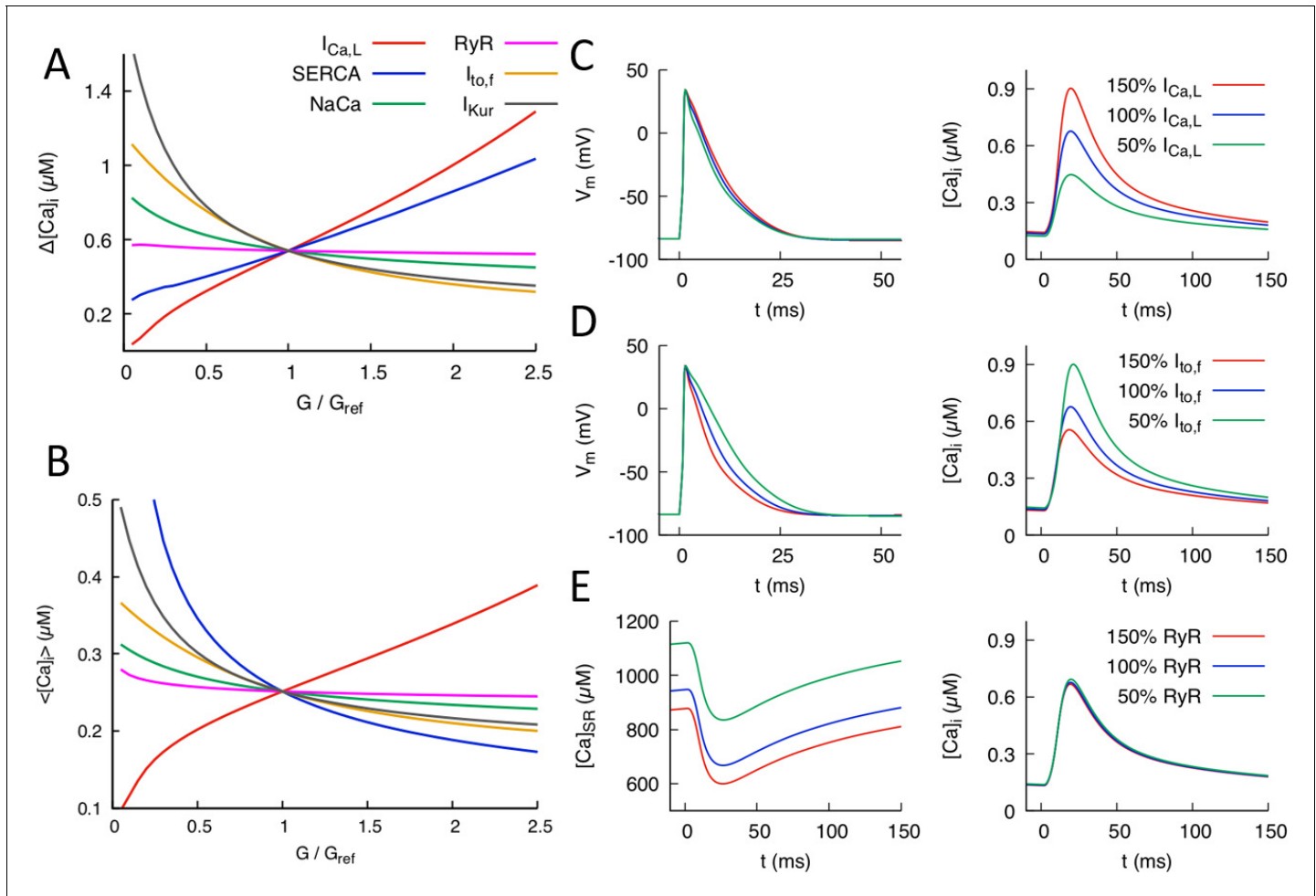

**Figure 2.** Effects of individual conductances on the $Ca^{2+}$ transient (CaT). (A) CaT amplitude defined as the difference $\Delta[Ca]_i$ between the peak and diastolic values of the cytosolic $Ca^{2+}$ concentration $[Ca]_i$ versus $G/G_{ref}$ where $G$ is the individual conductance value and $G_{ref}$ some fixed reference value. (B) Time-averaged $[Ca]_i$ over one pacing period ($\langle[Ca]_i\rangle$) versus $G/G_{ref}$. Illustration of the effect of varying $I_{Ca,L}$ conductance (C) and $I_{to,f}$ conductance (D) on AP and CaT profiles, where 50%, 100%, and 150% correspond to $G_{ref}$=0.5, 1.0, and 1.5, respectively. (E) Effect of varying RyR conductance on SR $Ca^{2+}$concentration $[Ca]_{SR}$ and CaT. Different time windows are plotted for the CaT and SR load (0 to 150 ms) and AP waveforms (0 to 50 ms) in (C—E).
DOI: https://doi.org/10.7554/eLife.36717.003

weak effect on the AP waveform (*Figure 2C*). The effect on the CaT stem from the fact that $I_{Ca,L}$ is the main trigger of $Ca^{2+}$-induced $Ca^{2+}$ release(CICR), which transfers a large amount of $Ca^{2+}$ from the SR to the cytosol. The weak effect on APD is due to the fact that the increase of CaT amplitude causes $I_{Ca,L}$ to inactivate more rapidly, thereby opposing AP prolongation. In contrast, increasing the conductances of $K^+$ currents that are dominant in the mouse such as $I_{to,f}$ (fast inactivating component of the transient outward current) and $I_{Kur}$ causes both $\Delta[Ca]_i$ and $\langle[Ca]_i\rangle$ to decrease. Increasing either of those $K^+$ currents speeds up repolarization (as illustrated for $I_{to,f}$ in *Figure 2D*) and hence inactivation of $I_{Ca,L}$, thereby reducing the magnitude of CICR. Increasing the conductance of the sodium-calcium exchanger current $I_{NaCa}$ also causes both $\Delta[Ca]_i$ and $\langle[Ca]_i\rangle$ to decrease by enhancing the forward mode of this current that extrudes $Ca^{2+}$ from the cytosol. The results of *Figure 2A* are consistent with a study of the effects of single conductance change in rat ventricular myocytes (*Devenyi and Sobie, 2016*), albeit with a much stronger influence of $I_{Ca,L}$ conductance on CaT amplitude in the present mouse model. Changing RyR expression from its reference value is seen to leave $\Delta[Ca]_i$ and $\langle[Ca]_i\rangle$ almost unchanged, even though it strongly affects the SR $Ca^{2+}$ concentration $[Ca]_{SR}$ (*Figure 2E*). This behavior reflects the well-known effect that making RyR channels more leaky (e.g. by addition of caffeine that increases RyR activity or, similarly here, by increasing the magnitude of the $Ca^{2+}$ release flux through RyRs) yields a transient increase in CaT amplitude, but no change in the steady-state CaT amplitude after $[Ca]_{SR}$ adjusts to a lower steady-state level (*Bers, 2001*). This effect is illustrated by time traces of $[Ca]_{SR}$ and $[Ca]_i$ in steady-state for different RyR expression levels in *Figure 2E*. Finally, changing the expression level of SERCA has opposite effects on $\Delta[Ca]_i$ and $\langle[Ca]_i\rangle$. Increasing SERCA magnitude increases SR $Ca^{2+}$ load, thereby increasing the amount of SR $Ca^{2+}$ release and CaT amplitude, but at the same time depletes $Ca^{2+}$ from the cytosol.

## Computationally determined good enough solutions

Next, we performed a computational search for combinations of parameters that yield a normal electrophysiological output as defined by the steady-state CaT amplitude $\Delta[Ca]_i$, time averaged cytosolic $Ca^{2+}$ concentration $\langle[Ca]_i\rangle$, and intracellular sodium concentration $[Na]_i$ at a 4 Hz pacing frequency. A GES search that uses the diastolic and peak $[Ca]_i$ values as $Ca^{2+}$ sensors, instead of $\Delta[Ca]_i$ (the difference between the peak and diastolic $[Ca]_i$ values) and time averaged $[Ca]_i$, gives nearly identical results. So our $Ca^{2+}$ sensors can be straightforwardly interpreted physiologically as requirements of normal diastolic and systolic contractile function necessary for a normal arterial blood pressure at the organism scale. A 'good enough solution' (GES) was defined as a combination of electrophysiological parameters that produces output values of those three quantities that are close enough to normal target values, which are defined as the values $\Delta[Ca]_i^*$, $\langle[Ca]_i\rangle^*$, and $[Na]_i^*$ corresponding to the reference set of parameters ($G_{ref}$ values) of the ventricular mycoyte model. The search was conducted by defining a cost function

$$E(\mathbf{p}) = \sqrt{\sum_{n=1}^{N}\left(\frac{S_n(\mathbf{p}) - S_n^*}{S_n^*}\right)^2} \leq \epsilon, \tag{1}$$

which is an aggregate measure of the deviation of output sensors $S_n(\mathbf{p})$ from their desired target values $S_n^*$. Here, $N=3$ with $S_1 = \Delta[Ca]_i$, $S_2 = \langle[Ca]_i\rangle$, and $S_3 = [Na]_i$, and $\epsilon$ is a small tolerance that we choose to be 5%. $E$ is a function of model parameters $\mathbf{p} = (p_1, p_2, \ldots)$ chosen to consist of the conductances of $I_{Ca,L}$, $I_{to,f}$, $I_{Kur}$, and $I_{NaCa}$ as well as RyR and SERCA expression levels. Effects of individual changes of those parameters on CaT properties measured by $S_1$ and $S_2$ are shown in *Figure 2A, B*. Conductances of other sarcolemmal currents that were found to have a negligible effect on the CaT were kept constant. The search for GES was conducted by first generating a large population of $\sim 10,000$ randomly chosen candidate models, with each model represented by a single parameter set $\mathbf{p}$. A candidate model was generated by randomly assigning each parameter $(p_1, p_2, \ldots)$ a value comprised between 0% and 300% of its reference value $G_{ref}$. We then utilized a multivariate minimization algorithm (see Materials and methods for details) that evolves $\mathbf{p}$ until the GES optimization constraint defined by *Equation 1* is satisfied. This method typically yields a large number of GES (7263 of the $\sim 10,000$ trials yield a GES with six parameters and three sensors described above, with

2737 either not converging or not producing a physiological output) and is computationally more efficient than a random search without optimization that yields very few GES.

Results of the GES search are shown in *Figure 3*. *Figure 3A* shows the parameters of six representative GES and their corresponding AP waveforms (*Figure 3B*) and CaT profiles (*Figure 3C*). The CaT profiles are all very close to each other, which holds for all GES, while the AP waveforms exhibit larger variations owing to the fact that the GES search does not involve any voltage sensing. *Figure 3D* shows histograms of parameters for all GES. Conductances of sarcolemmal currents tend to be highly variable except for $I_{NaCa}$, which turns out to be constrained by the intracellular sodium concentration sensor ($S_3 = [Na]_i$). This is revealed by a GES search with only $Ca^{2+}$ sensing ($S_1$ and $S_2$) that yields a broader histogram for the $I_{NaCa}$ conductance (*Figure 3—figure supplement 1*). The histogram of RyR expression level is very broad. This is consistent with the fact that this parameter was found to have a very weak effect on the CaT (*Figure 2A,B*) due to the compensatory adjustment of SR $Ca^{2+}$ load (*Figure 2E*). In contrast, the histogram of SERCA is very narrow. This feature, which persists even if $Na^+$ sensing is removed (*Figure 3—figure supplement 1*), is predominantly due to $Ca^{2+}$ sensing. It stems from the fact that changing SERCA expression level has opposite effects on the CaT amplitude (*Figure 2A*) and average $[Ca]_i$ (*Figure 2B*), increasing one while decreasing the other or vice-versa. Therefore, those opposite effects cannot be compensated by changes of conductance of sarcolemmal currents that simultaneously increase or decrease both $Ca^{2+}$ sensors, or by changes of RyR expression level that has a negligible effect on the CaT due to SR load adjustment. However, conductances of inward and outward currents that change both $Ca^{2+}$ sensors in opposite directions can in principle compensate each other. This compensation is revealed by representing each GES as a point in a 3D plot (*Figure 3E*) whose axes are the conductances of $I_{Ca,L}$, $I_{to,f}$ and $I_{Kur}$. This plot shows that all GES lie close to a 2D surface in this 3D conductance space due to a three-way compensation between the effects of $I_{Ca,L}$, $I_{to,f}$, and $I_{Kur}$ on the CaT. GES lie inside a smeared 2D surface (i.e. a 2D surface of finite thickness) in the 3D conductance space of *Figure 3E*. This feature stems from the fact that the GES parameter space considered here is in principle six-dimensional (four sarcolemmal current conductances and 2 $Ca^{2+}$ protein expression levels). However, SERCA expression and $I_{NaCa}$ conductance are constrained by $Ca^{2+}$ and $Na^+$ sensing, respectively, and RyR expression has a negligible effect on both $Ca^{2+}$ and $Na^+$ sensors, thereby reducing the relevant parameter space to the three conductance axes of *Figure 3C*. The subspace of GES that minimizes the cost function $E$ must therefore lie on the 2D surface $E = 0$. This surface is smeared because $I_{NaCa}$ is only constrained by $Na^+$ sensing within a finite range and the GES search only minimizes $E$ within a finite tolerance ($E \leq \epsilon$ instead of $E = 0$).

To facilitate the comparison with experiments presented in the next subsection, it is useful to represent the three-way compensation between $I_{Ca,L}$, $I_{to,f}$, and $I_{Kur}$ conductances by plotting the sum of the peak currents of $I_{to,f}$ and $I_{Kur}$ versus the peak current of $I_{Ca,L}$ with all three currents measured under voltage-clamp with a step from $-50$ to $0$ mV. Those peak currents are proportional to conductances up to proportionality factors fixed by intra- and extracellular ionic concentrations and voltage. In this peak-current representation, the smeared 2D surface of GES of *Figure 3E* takes on the simpler form of a thick nearly straight line (*Figure 3F*). We note that even though correlations between two or more parameters have been explored in population models (*Sánchez et al., 2014*; *Britton et al., 2013*; *Muszkiewicz et al., 2018*), their sum studied here has not been previously considered.

## Good enough solutions in the HMDP

In order to test the computational modeling predictions, and at the same time differentiate intra-heart cell-to-cell from inter-subject variability, we performed electrophysiological and contractile measurements on ventricular myocytes obtained from mouse hearts of nine different strains from the HMDP listed in the Materials and methods, each strain assumed to represent a different good enough solution. Peak values of $I_{Ca,L}$, $I_{to,f}$, and $I_{Kur}$ were measured under voltage-clamp with a step from $-50$ to $0$ mV following established protocols (see Materials and methods). The $K^+$ currents were measured in the same cell and the $Ca^{2+}$ currents in different cells. Contraction was analyzed by measuring mycoyte shortening during several paced beats in separate cells for six strains that include five of the strains in which conductances were measured. In order to collect enough statistics to distinguish cell-to-cell from inter-strain variability, several hearts of each isogenic strain were

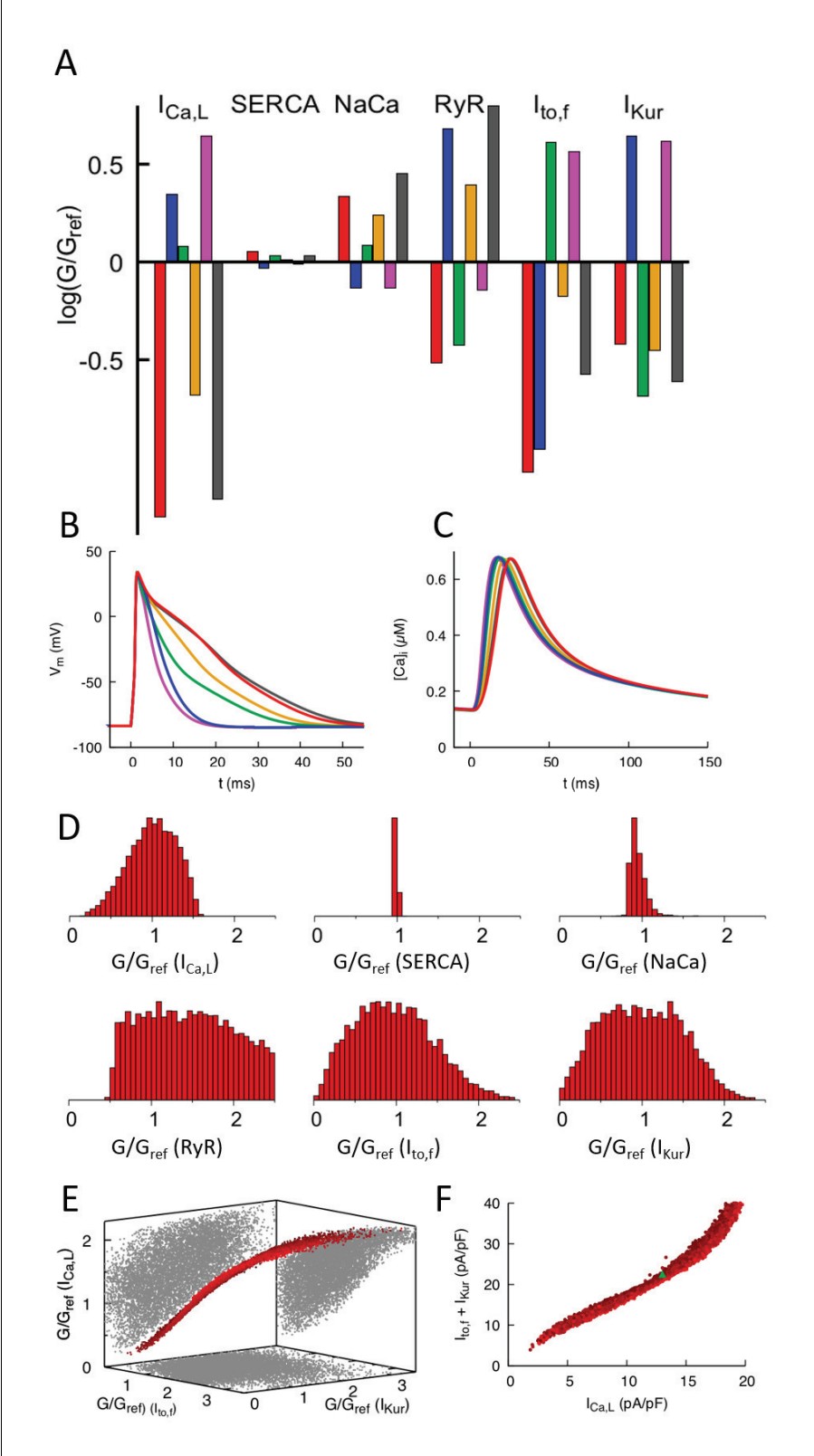

**Figure 3.** Computationally determined good enough solutions (GES) with calcium sensing. (**A**) Examples of GES representing combinations of 6 conductances that produce a normal CaT and intracellular Na$^+$ concentration. Each color represents a different GES and the corresponding AP and CaT profiles are shown in B) and C), respectively. (**D**) Histograms of individual normalized conductances G/G$_{ref}$ for a collection of 7263 GES showing that some conductances are highly variable while others are highly constrained. (**E**) Three-dimensional (3D) plot revealing a three-way compensation

*Figure 3 continued on next page*

*Figure 3 continued*

between conductances of $I_{Ca,L}$, $I_{to,f}$, and $I_{Kur}$. Each GES is represented by a red dot. All GES lie close to a 2D surface in this 3D plot. Pairwise projections (grey shadows) do not show evidence of two-way compensation between pairs of conductances. (F) Alternate representation of three-way compensation obtained by plotting $I_{Ca,L}$ versus the sum of $I_{to,f}$ and $I_{Kur}$. Peak values of those currents after a voltage step from −50 to 0 mV are used to make this plot that can be readily compared to experiment. Different time windows are plotted for the AP waveforms and CaT in B and C, respectively.

DOI: https://doi.org/10.7554/eLife.36717.004

The following figure supplements are available for figure 3:

**Figure supplement 1.** Histograms of individual ion channel conductances in 8320 GESs found by a GES search constrained only by $Ca^{2+}$ transient amplitude and average, but not constrained by intracellular sodium concentration $[Na]_i$.

DOI: https://doi.org/10.7554/eLife.36717.005

**Figure supplement 2.** Correlation between $I_{Ca,L}$ and the sum of $I_{to,f}$ and $I_{Kur}$ is weaker but still significant when intracellular sodium concentration is not constrained.

DOI: https://doi.org/10.7554/eLife.36717.006

**Figure supplement 3.** Computationally determined GES with voltage sensing.

DOI: https://doi.org/10.7554/eLife.36717.007

used. The number of cells that could be obtained from one heart for L-type $Ca^{2+}$ current, several $K^+$ currents, or contraction analysis varied from 1 to 7 so that several hearts of each strain were needed to obtain enough independent measurements to statistically distinguish intra-heart cell-to-cell from inter-strain variability (see data in Materials and methods).

The results of current and contraction measurements are shown in *Figure 4*. In *Figure 4A*, we plot the sum of the peak currents of $I_{to,f}$ and $I_{Kur}$ versus the peak current of $I_{Ca,L}$ together with the standard errors of the mean (SEM) of those quantities. Bar plots showing mean current values together with both SEM and standard deviation (SD) characterizing cell-to-cell variability are given in the Materials and methods. We also superimpose on this plot the computationally predicted GES of *Figure 3F*. Different HMDP strains, each representing a GES, are seen to function with different combinations of $Ca^{2+}$ and $K^+$ currents that compensate each other in a non-trivial three-way fashion that closely follows the GES computationally determined with a three-sensor search in which both the CaT and $Na^+$ concentration are constrained (faded red points in *Figure 4A*). The sum of $I_{to,f}$ and $I_{Kur}$ follows a linear correlation with $I_{Ca,L}$ (p=0.0007) using eight out of nine strains and the correlation remains statistically significant (p=0.0144) if the outlier strain (BXA12/PgnJ) is included. Interestingly, this outlier strain still falls within the larger range of computationally predicted GES using a two-sensor search without $Na^+$ sensing (faded blue points in *Figure 4A*). To distinguish cell-to-cell from inter-strain variability, we performed a one-way ANOVA F-test on the $I_{Ca,L}$ measurements. The result shows that $I_{Ca,L}$ measurements for all strains do not originate from the same distribution (p-value p=0.000024). Furthermore, we performed a student T-test using raw data of $I_{Ca,L}$ measurements for all pairs of strains. The results yield very small statistically significant p-values for pairs of strains with sufficiently different average current values (e.g. BXA25/PgnJ, CXB1/ByJ, and C57BL/6J in *Figure 4A*). Those results are consistent with the fact that mean currents differ much more than their standard error for those strains, as can be seen by visual inspection of means and SEM values corresponding to thin bars on both axes of *Figure 4A*. We conclude that inter-strain variability of ion channel conductances can be distinguished from cell-to-cell variability of those same quantities for a significant number of the strains investigated. While the $Ca^{2+}$ and $K^+$ currents were measured for the nine strains reported in *Figure 4A*, the $Ca^{2+}$ current was measured in seven additional strains (total of 16 strains). Those additional measurements reported in the Materials and methods confirm that some strains can have markedly different $I_{Ca,L}$ conductances.

Unlike ion channel conductances, CaT properties were assumed not to vary in the computationally-enabled GES search, which rests on the hypothesis that $Ca^{2+}$ sensing provides a feedback mechanism that regulates ion channel gene and protein expression. The results in *Figure 4B*, which use contraction as a surrogate for CaT amplitude, support this hypothesis by showing that mean values of cell shortening do not vary substantially across strains. This is confirmed by performing a standard ANOVA statistical test, which shows that cell shortening measurements for the six strains reported in *Figure 4B* do not originate from different distributions within statistical uncertainty (p-value p=0.4136).

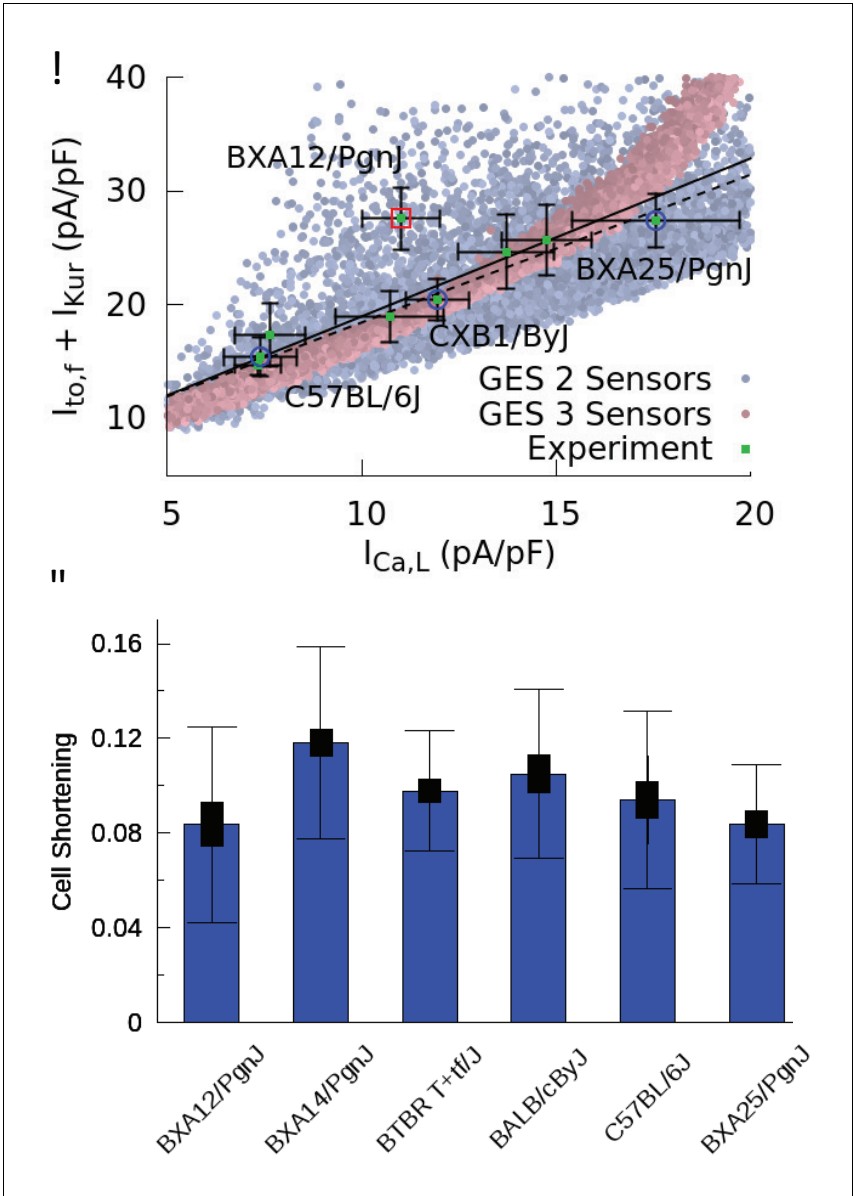

**Figure 4.** Good enough solutions in the Hybrid Mouse Diversity Panel (HMDP). (**A**) Central result of this paper showing quantitative agreement between theoretically predicted and experimentally measured compensation of inward $Ca^{2+}$ and outward $K^+$ currents. Equivalent plot of *Figure 3F* showing the sum of $I_{to,f}$ and $I_{Kur}$ versus $I_{Ca,L}$ for nine different mouse strains using peak values of those currents (proportional to conductances) after a voltage step from $-50$ to $0$ mV. Mean current values (green filled squares) are shown together with standard errors of the mean (thin bars) for each strain. The number of cells used for each strain is given in *Table 1* of the Materials and methods section. Computationally determined GES are superimposed and shown as faded red points using all three sensors (CaT amplitude, average $[Ca]_i$, and diastolic $[Na]_i$) and faded blue points for two sensors (CaT amplitude and average $[Ca]_i$). Lines represent linear regression fits using the method of Chi-squared minimization with errors in both coordinates including (solid line, p=0.0144) and excluding (dashed line, p=0.0007) the outlier strain BXA12/PgnJ marked by a red box. The small p values of those fit validate the computationally predicted three-way compensation of $Ca^{2+}$ and $K^+$ currents. The three strains selected for the organ scale study (C57BL/6J, CXB1/ByJ, and BXA25/PgnJ) with low, medium, and high $I_{Ca,L}$ conductance, respectively, are highlighted by blue circles. (**B**) Cell shortening, measured as the fraction of resting cell length at 4 Hz pacing frequency in different HMPD strains where thick and thin bars correspond to standard error of the mean and standard deviation, respectively. A standard ANOVA test shows no significant differences in cell shortening between strains (p=0.4136) supporting the hypothesis that different combinations of conductances produce a similar CaT and contractile activity.

*Figure 4 continued on next page*

*Figure 4 continued*

DOI: https://doi.org/10.7554/eLife.36717.008

The following figure supplement is available for figure 4:

**Figure supplement 1.** Patch clamp measurements of mean $I_{Ca,L}$ (A), $I_{to,f}$ (B), and $I_{Kur}$ (C) functional current density averaged over multiple cells for nine HMDP mouse strains with standard errors (thick bars) and standard deviations (thin bars).

DOI: https://doi.org/10.7554/eLife.36717.009

## Compensation at the organ scale

Current measurements discussed in the previous section show that mean conductances of $Ca^{2+}$ and $K^+$ currents vary between strains in a compensatory way so as to produce a normal CaT. They also show that conductances vary significantly from cell to cell around their mean values. This is illustrated in *Figure 5A* for three mouse strains that have statistically distinguishable mean $I_{Ca,L}$ conductances (low, medium, and high) as measured by standard errors (thick bars), but exhibit large cell-to-cell variability as measured by the standard deviations (thin bars) of the distributions of conductance measurements in individual cells. This raises the question of whether compensation remains operative at the organ scale in the presence of large cell-to-cell variability. There are two interlinked aspects to this question. The first relates to the cellular-level dynamical coupling between membrane voltage and intracellular $Ca^{2+}$ dynamics that is inherently nonlinear (*Krogh-Madsen and Christini, 2012*; *Karma, 2013*; *Qu et al., 2014*). Even when cells are uncoupled, this nonlinearity could potentially cause the mean CaT amplitude in an ensemble of cells with highly variable conductances to differ from the CaT amplitude computed in a single cell with conductances set to the mean values of the ensemble, as traditionally done in cardiac modeling. The second aspect relates to the additional effect of gap-junctional coupling between cells. This effect is well-known to smooth out cell-to-cell variation of AP waveforms on a mm scale that is much larger than the individual mycoyte length. However, whether this smoothing translates into increased organ-scale uniformity of CaT amplitude and contractility is unclear.

To address those two aspects, we constructed tissue scale computational models for three mouse strains with statistically distinguishable average conductances (as illustrated for $I_{Ca,L}$ in *Figure 5A*). Tissues of each strain consisted of $56 \times 56$ electrically coupled cells (see Materials and methods for details). Simulations were carried out with and without electrical coupling to assess the effect of the latter. The conductances of $I_{Ca,L}$, $I_{to,f}$, and $I_{Kur}$ were assumed to vary randomly from cell to cell, and no constraint was imposed on the ratio of $I_{to,f} + I_{Kur}$ to $I_{Ca,L}$, representing the worst case scenario in which both AP and CaT would exhibit maximal variations at the single myocyte level. Their values were drawn randomly from Gaussian distributions with average values and standard deviations that match experimental current measurements in each strain. All other parameters were kept fixed to reference values. The resulting cell-to-cell variation of conductances for three different strains is shown in *Figure 5B*) using the same peak-current representation of *Figures 3F* and *4A*. In this representation, each point represents a different cell, and clouds of points of the same color represent all cells in a tissue of the same strain. Furthermore, the center of each cloud falls on the thick line corresponding to the computationally determined GES surface where compensation is operative at the single-cell level.

The results of simulations with populations of uncoupled and coupled cells with randomly varying conductances are shown in *Figure 5C–G*. *Figure 5C* shows that AP waveforms are highly variable when cells are uncoupled, reflecting the variability in conductances with no constraint imposed on the ratio of $I_{to,f} + I_{Kur}$ to $I_{Ca,L}$. *Figure 5D* shows that AP waveforms becomes uniform when cells are coupled, as expected, even though interstrain variability is still significant. *Figure 5E* compares histograms of CaT amplitude and AP duration (APD) when cells are uncoupled and coupled. Consistent with the AP waveforms of *Figure 5C and D*, APD histograms in *Figure 5E* show that junctional coupling strongly reduces APD variability, as expected. CaT amplitude and average $[Ca]_i$ histograms in turn reveal that, in coupled cells, the more uniform APD translates into a more uniform CaT amplitude and average $[Ca]_i$ (i.e. narrower ΔCa and <Ca> histograms, respectively), reflecting the influence of the cell's APD on its CaT. At the organ scale, this ensures that cells in tissue have uniform APs. They also benefit modestly from a more uniform CaT as a result of coupling, promoting more

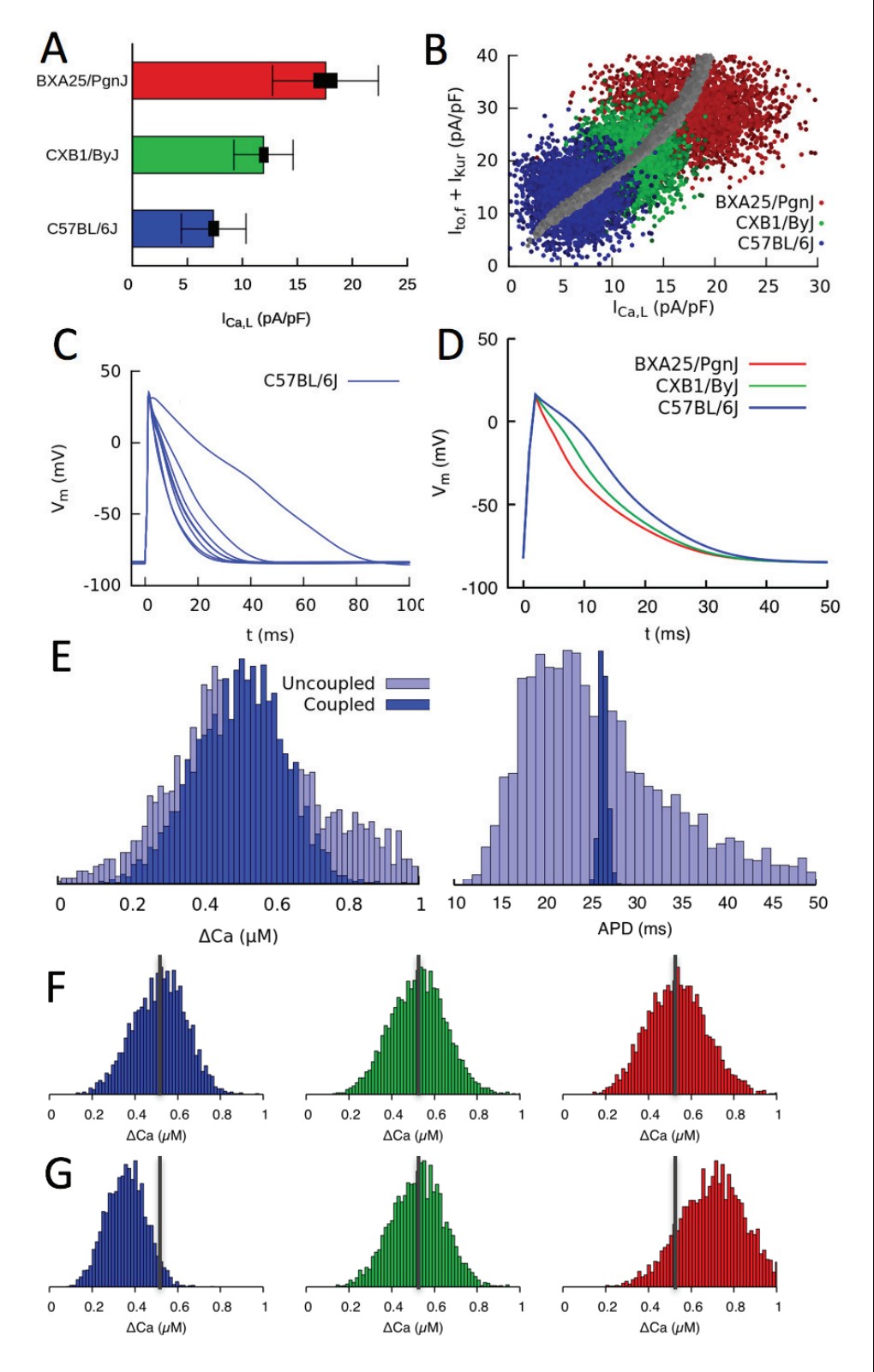

**Figure 5.** Organ scale compensation. (**A**) Mean $I_{Ca,L}$ conductance in three different HMDP strains where thick and thin bars denote standard error and standard deviation, respectively. (**B**) Sets of conductances generated to be representative of individual cells within ventricular tissue of the three strains by assigning normally distributed random values to the $I_{Ca,L}$, $I_{to,f}$ and $I_{Kur}$ conductances using experimentally determined means and standard deviations. The blue, green, and red points correspond to the three HMDP strains with low (C57BL/6J), medium (CXB1/ByJ), and high (BXA25/PgnJ)

*Figure 5 continued on next page*

*Figure 5 continued*

$I_{Ca,L}$ conductance, respectively, and the grey points are the results of the three-sensor GES search (same as *Figure 3F*). (C) Variable AP waveforms in uncoupled myocytes with conductances randomly chosen from the distribution shown in B for C57BL/6J and D) AP waveforms for coupled myocytes in tissue for C57BL/6J and the two other strains. AP waveforms of uncoupled cells vary significantly from cell to cell as observed experimentally (Fig. *Figure 5—figure supplement 1*) but are uniform in electrotonically coupled cells, as expected. (E) Histograms of $Ca^{2+}$ transient (CaT) amplitude ($\Delta Ca$) and action potential duration (APD) for C57BL/6J in electrotonically uncoupled and coupled cells. Importantly, in coupled cells, the more uniform APD translates into a much more uniform CaT amplitude, reflecting the strong effect of the cell's APD on its CaT amplitude. (F) Distribution of CaT amplitudes within electrotonically coupled cells in tissue scale simulations using the parameter distributions from B. The three strains have the same mean CaT amplitude averaged over all cells marked by a thick vertical gray line, thereby demonstrating that compensation of $Ca^{2+}$ and $K^+$ currents remains operative at a tissue scale. (G) Distribution of CaT amplitudes obtained by varying only $I_{Ca,L}$ conductance and with $I_{to,f}$ and $I_{Kur}$ conductances fixed to their reference values. Lack of compensation between $Ca^{2+}$ and $K^+$ currents in this case yields different mean CaT amplitude.

DOI: https://doi.org/10.7554/eLife.36717.011

The following figure supplements are available for figure 5:

**Figure supplement 1.** Action potential recordings from isolated myocytes for mouse strain C57BL/6J paced at 4 Hz under current clamp.
DOI: https://doi.org/10.7554/eLife.36717.012

**Figure supplement 2.** Histogram of average $Ca^{2+}$ concentration corresponding to *Figure 5E* for C57BL/6J in electrotonically uncoupled and coupled cells.
DOI: https://doi.org/10.7554/eLife.36717.013

uniform force generation throughout the tissue. Thus, tissue coupling compensates significantly when AP and CaT variability between single myocytes is high, for the case in which the ratio of $I_{to,f} + I_{Kur}$ to $I_{Ca,L}$ is not constrained at the single myocyte level.

*Figure 5F and G* show that compensation remain operative at the organ scale. *Figure 5F* shows that, even though the strains have different average conductances (*Figure 5B*), they produce CaT amplitude histograms with approximately the same mean and width. In contrast, if the same simulation is repeated by fixing the conductance of $K^+$ currents to the value of the strain with the medium value of $I_{Ca,L}$ conductance (CXB1/ByJ), the different $I_{Ca,L}$ conductances are not compensated by different $I_{to,f}$ and $I_{Kur}$ conductances, yielding CaT amplitude distributions with shifted peaks and hence different aggregate contraction (*Figure 5G*).

In summary, our results show that compensation remains operative at the organ scale because CaT amplitude histograms have similar means with and without electrical coupling (*Figure 5E*). This implies that cell-to-cell variability of conductances and hence APD causes variability of CaT amplitude without significantly affecting its mean, so that $I_{Ca,L}$ and potassium currents can compensate each other even though conductances exhibit large cell-to-cell variations from their mean values. Gap junctional coupling has the additional important effect of reducing CaT amplitude variability, thereby promoting tight organ-level behavior despite high cell-to-cell variability.

## Cardiac hypertrophic response to a stressor

From a functional standpoint, the most relevant implication of the present study is that different GES may exhibit markedly different responses to perturbations, as previously demonstrated in a neuroscience context (*Grashow et al., 2009*). To examine this possibility, we reviewed data from separate studies of isoproterenol (ISO)-induced cardiac hypertrophy and heart failure in approximately 100 HMDP strains that include most of the strains used in the present study. In those studies, heart mass was measured in those strains before ($m_{pre}$) and 3 weeks after ($m_{post}$) implantation of a pump continuously delivering isoproterenol (*Table 3*). *Figure 6* reveals the existence of a statistically very significant correlation between baseline $I_{Ca,L}$ conductance and the hypertrophic response ($m_{post}/m_{pre}$). Although many factors contribute to the hypertrophic response in the HMDP (*Rau et al., 2017*; *Santolini et al., 2018*), intracellular $Ca^{2+}$ overload activating the $Ca^{2+}$-calcineurin-NFAT signaling pathway has been shown to play a major role (*Bers, 2008*). Since $I_{Ca,L}$ is the major pathway of $Ca^{2+}$ entry into the cytoplasm, it is intriguing to speculate that strains with a larger baseline $I_{Ca,L}$ conductance under baseline conditions have a more robust increase in $I_{Ca,L}$ that is not adequately compensated by repolarizing $K^+$ currents, making those strains more susceptible to $Ca^{2+}$ overload when $I_{Ca,L}$ is enhanced during sustained $\beta$-adrenergic stimulation by isoproterenol. Hypothetically, this may result in a stronger cardiac hypertrophic response. To make this case convincingly, however, would require demonstrating that $Ca^{2+}$ overload is chronically worsened in strains

with a high baseline $I_{Ca,L}$ conductance and ruling out other strain-dependent hypertrophy-promoting pathways that are not $Ca^{2+}$-dependent, which is beyond the scope of the present work.

## Compensation and gene expression

In a neuroscience context, ionic conductances of neurons from the stomatogastric ganglion of different crabs were previously found by *Schulz et al. (2006)* to be correlated with gene expression, as shown by independent measurements in the same subjects of functional densities of different ion channels, used to determine conductances, and mRNA levels of genes encoding for pore-forming subunits of those channels. In a cardiac context, decrease of $I_{Ca,L}$ current density has been shown to be correlated with a decrease of Cav1.2 mRNA expression in response to a sustained increase of pacing rate in cultured adult canine atrial cardiomyocytes mimicking atrial tachycardia remodeling (*Qi et al., 2008*). In the present study, we did not perform independent measurements of gene expression in the same ventricular myocytes used to measure ionic conductances. However, to examine the possible relationship between compensation of conductances and gene expression, we reviewed the gene expression data from the aforementioned studies of ISO-induced cardiac hypertrophy and heart failure in approximately 100 HMDP strains that include most of the strains used in the present study. Gene expression was measured both in control (pre-ISO) and after injection of ISO for 21 days in 8- to 10-week-old female mice (post-ISO). Details of heart biopsies conducted pre- and post-ISO and microarray data analysis are given in the Methods section of *Santolini et al. (2018)*.

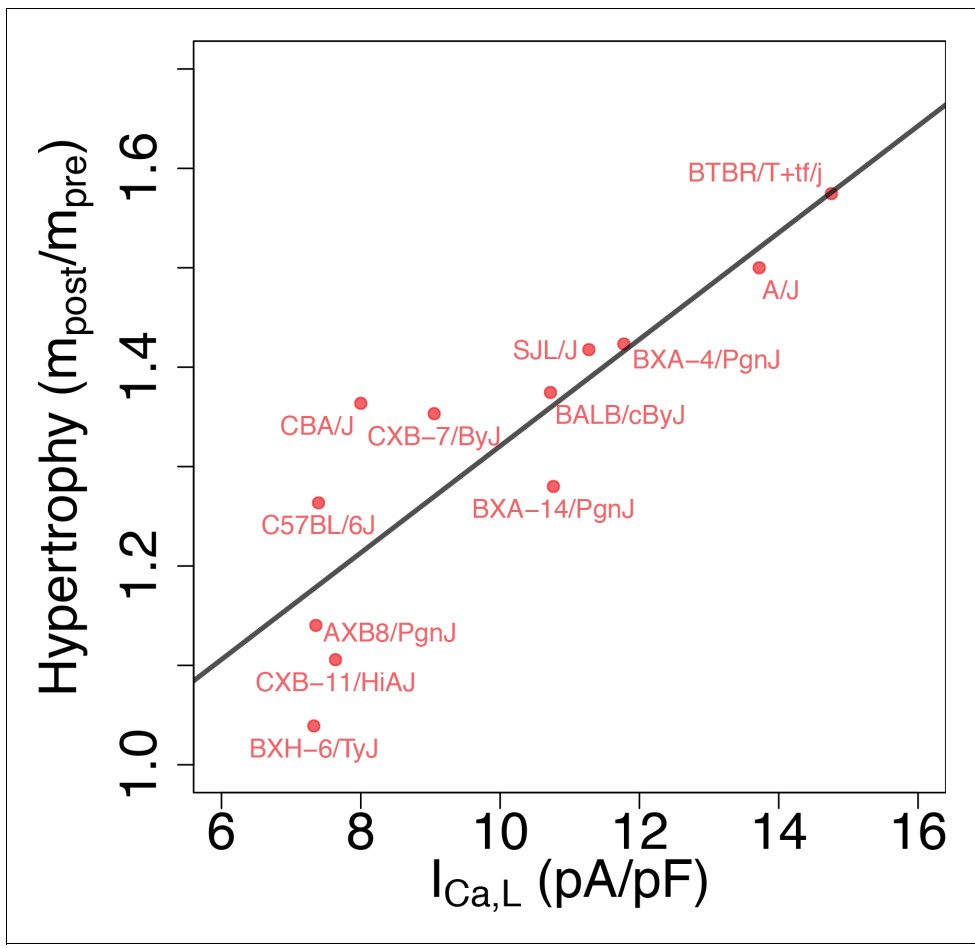

**Figure 6.** Correlation between L-type $Ca^{2+}$ current conductance and cardiac hypertrophic response to a stressor for different HMDP strains. The Pearson correlation is r = 0.86 (p=3e-4).
DOI: https://doi.org/10.7554/eLife.36717.014

No statistically significant pairwise correlation (Pearson correlation coefficient $r<0.25$ and p-value $p>0.05$) were found between expression levels of the genes Cacna1c, Kcnd2, and Kcna5, encoding for the pore forming subunit of the Cav1.2, Kv4.2, and Kv1.5 channels associated with $I_{Ca,L}$, $I_{to,f}$, and $I_{Kur}$, respectively. However, a statistically very significant correlation was found between expression levels of Cacna1c and Kcnip2 that encodes the KChIP2 accessory $\beta$ subunits directly interacting with Kv4.2 (see *Figure 7* and caption for $r$ and $p$ values). This correlation is present both in control (pre-ISO), which is the condition relevant to our conductance measurements in selected strains (*Figure 4*), and post-ISO. Since increased KChIP2 level is known to increase the functional current density of $I_{to,f}$ (*Kuo et al., 2001*; *Jin et al., 2010*), the strong positive correlation between Cacna1c (Cav1.2) and Kcnip2 (KChIP2) expression levels may partially contribute to the positive correlation between $I_{Ca,L}$ and $I_{to,f}+I_{Kur}$ functional current densities (*Figure 4*).

## Discussion

In the present study, we have proposed a new methodology for searching for combinations of electrophysiological parameters representing different individuals in a genetically diverse population. While previous studies have used primarily AP features to constrain parameters (*Sarkar and Sobie, 2009*; *Sarkar and Sobie, 2010*; *Sarkar et al., 2012*; *Britton et al., 2013*; *Groenendaal et al., 2015*; *Muszkiewicz et al., 2016*; *Krogh-Madsen et al., 2016*), we have chosen to constrain parameters using the $Ca^{2+}$ transient that plays a key role to regulate ion channel expression and activity. This choice is based on a straightforward physiological hypothesis, namely that the CaT is critical for generating blood pressure, which is sensed by the carotid baroreceptors and feeds back through the autonomic nervous system to regulate the CaT in a way that preserves blood pressure. In contrast, a physiological basis for sensing cardiac voltage to regulate the AP and CaT is unclear. We have also examined the effect of additionally constraining the intracellular $Na^+$ concentration that is also known to modulate ion channel activity. Regulatory mechanisms traverse different levels of biological organization from transcriptional regulation to post-transcriptional and post-translational modification to ion channel trafficking and phosphorylation. Those mechanisms are presently not known in sufficient detail to be modeled quantitatively. However, there is sufficient experimental evidence of feedback sensing of cellular activity via $Ca^{2+}$ (*Qi et al., 2008*) and $Na^+$ (*Harvey et al., 1991*; *Balke and Wier, 1992*) concentrations to make a search that constrains model parameters based on those signals plausible. The $Ca^{2+}$ transient determines the contractile force underlying arterial blood pressure generation regulated by baroreceptor feedback via the autonomic nervous system. Hence, fixing the diastolic and peak $[Ca]_i$ values is a physiologically meaningful choice to search for parameter combinations that produce a normal diastolic and systolic function, which we have adopted here. Previous work (*Xiao et al., 2008*) has provided evidence of a compensatory increase of $I_{Ks}$ following exposure of canine cardiomyocytes to a pharmacological $I_{Kr}$ blocker. Even though the mechanisms are not clear, it has been hypothesized that feedback sensing of $[Ca]_i$ may potentially underlie the compensatory upregulation of $I_{Ks}$ through post-transcriptional upregulation of underlying channel subunits mediated by microRNA changes. Together with other studies (*Qi et al., 2008*), those previous findings may provide supportive evidence for the present $Ca^{2+}$ sensing hypothesis and suggests its generality beyond mice.

A remarkable and nontrivial finding of the present computational study is that $Ca^{2+}$ sensing suffices to produce a physiological AP waveform whose duration spans comparable range (see *Figure 5C*) to that recorded experimentally in isolated myocytes (*Figure 5—figure supplement 1*), even though the voltage signal is not used to constrain model parameters. For comparison, we show in *Figure 3—figure supplement 3* the results of a GES search that uses voltage instead of $Ca^{2+}$ sensing. With voltage sensing alone (both without and with $[Na]_i$ sensing), the AP waveform was readily constrained as expected, but the CaT was highly variable and often not physiological. Moreover, the correlation between inward $Ca^{2+}$ and outward $K^+$ currents observed in the HMDP (*Figure 4A*) was no longer preserved, since other inward and outward currents including $I_{NaCa}$ could regulate AP duration when the CaT was not constrained. Those results support our hypothesis that ionic conductances are primarily regulated by feedback mechanisms sensing ionic concentrations. Since the CaT is critical for generating blood pressure (which is sensed by the carotid baroreceptors and feeds back through the autonomic nervous system to regulate the CaT by controlling levels of Ca-cyling proteins and the AP in a way that preserves blood pressure), $Ca^{2+}$ sensing provides a

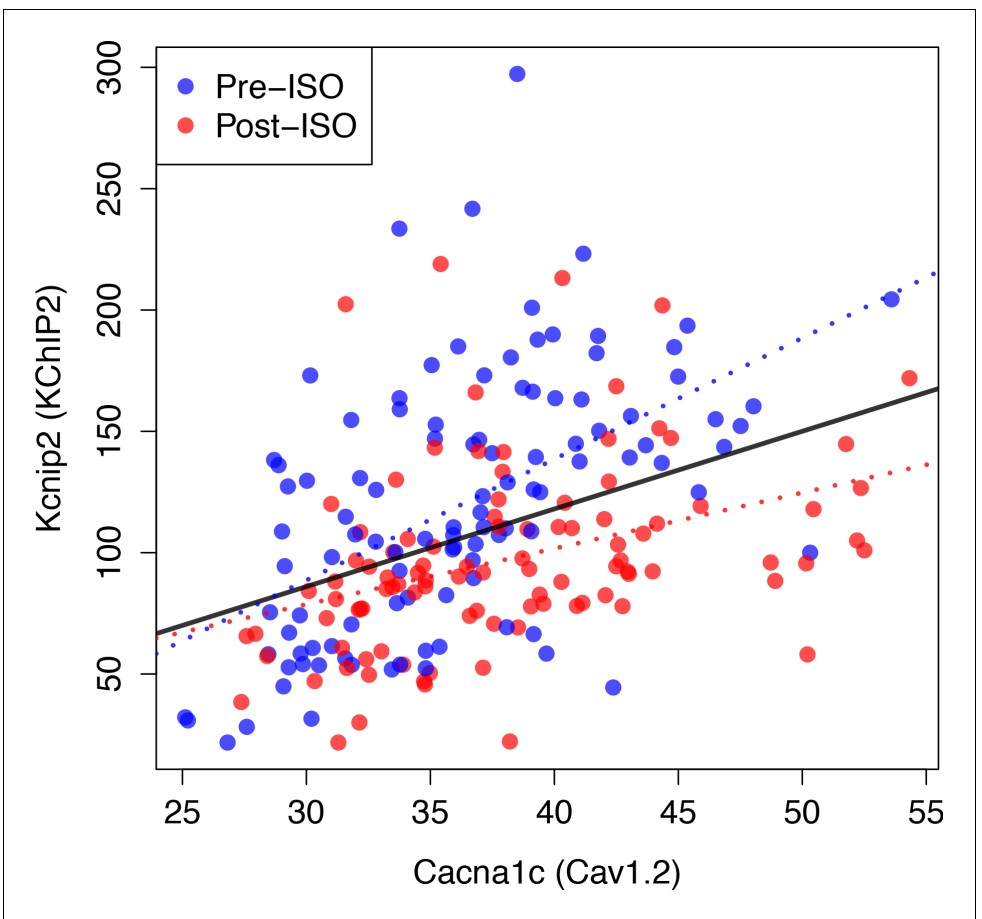

**Figure 7.** Compensation and gene expression. Plot showing the existence of a statistically very significant correlation (Pearson correlation coefficient $r = 0.47$ and p-value, $p = 8.1 \, 10^{-13}$) between the expression level of Kcnip2, encoding the KChIP2 accessory $\beta$ subunits that interact with Kv4.2 channels ($I_{to,f}$) and of Cacna1c, a gene encoding the $\alpha$1C subunit of the Cav1.2 L-type calcium channels ($I_{Ca,L}$) across 206 mice. Cardiac gene expression was measured in 106 control (Pre-ISO) strains and 21 days after injection of isoproterenol (post-ISO) in 100 HMDP strains (a smaller number due to higher mortality of certain strains). Note that the significant correlation holds when considering separately pre-ISO (blue points, $r = 0.59$, $p = 2 \, 10^{-11}$) and post-ISO (red points, $r = 0.42$, $p = 1.5 \, 10^{-5}$) data. Lines show best fits of a linear model for pre-ISO (blue), post-ISO (red), and pre- and post-ISO combined (black). Expression data is taken from *Santolini et al. (2018)* and is averaged over all microarray probes for each gene.

DOI: https://doi.org/10.7554/eLife.36717.016

straightforward physiological mechanism that not only constrains the CaT to a physiological waveform, but also, as an added and novel bonus, constrains AP features through the ratio of inward $Ca^{2+}$ current and outward $K^+$ currents. It is much less clear, on the other hand, how voltage would be sensed by the heart to provide a feedback mechanism to control both the AP waveform and the CaT. Voltage-sensing alone provided a reliable AP waveform, but a highly unreliable CaT as shown in *Figure 3—figure supplement 3*.

Our results show that CaT calibration results in considerable AP waveform variability (*Figure 5C*) in isolated myocytes as expected if the AP waveform is not constrained. This finding is consistent with the observations that AP variability is considerable when measured experimentally in patch clamp studies (*Figure 5—figure supplement 1*), but greatly reduced in tissue because less frequent atypical AP waveforms are voltage-clamped by the more typical AP waveforms of their neighbors.

Our finding that CaT calibration results in considerable AP waveform variability is also consistent with the converse finding in a previous study (*Muszkiewicz et al., 2018*) and here (*Figure 3—figure supplement 3*) that the CaT is highly variable when the AP waveform alone is constrained without

also constraining the CaT. However, at least in the present study, constraining the AP waveform does not reproduce the correlation between $Ca^{2+}$ and $K^+$ currents observed in the HMDP.

Even though our calcium centric GES search did not include the conductance of the $Na^+$ current, it seems physiologically plausible that this conductance (and gap junction coupling) could also be regulated by $Ca^{2+}$ sensing to ensure that conduction velocity is adequate to generate a synchronous blood pressure waveform. In particular, the integrated CaT of the ventricles has to be reasonably synchronous to generate a normal blood pressure waveform, requiring the Na current density to be adequate for a normal conduction velocity through the tissue.

Even though the GES search was performed using target values of $Ca^{2+}$ sensors for a 4 Hz pacing frequency, different GES exhibit similar CaT amplitude versus pacing frequency curves consistent with experimental measurements (Figure 18 in *Bondarenko et al., 2004*) over a broad range of pacing frequencies from 0.5 to 4 Hz. Since different model parameters corresponding to different strains reproduce similar CaT amplitude-frequency curves, we do not expect the choice of pacing frequency to be critically important for calibrating model parameters that produce a normal electrophysiological phenotype. We attribute the robustness of those curves to the knock on effect of voltage on the L-type $Ca^{2+}$ current and SR $Ca^{2+}$ release via CICR. As a result of this effect, constraining the CaT indirectly constrains the relative magnitudes of depolarizing and repolarizing currents affecting the AP; that is, the same CaT amplitude can be obtained with combinations of $Ca^{2+}$ and $K^+$ currents that are both large or both small, thereby compensating each other, but not with combinations in which the $Ca^{2+}$ current is large and the sum of $K^+$ currents is small or vice-versa. It remains that the AP waveform and duration are only partially constrained by the CaT and are thus more variable than in a GES search that uses AP features such as duration, plateau voltage, etc., to constrain parameter sets (*Sarkar and Sobie, 2009*; *Sarkar and Sobie, 2010*).

While the additional constraint to keep the intracellular $Na^+$ concentration $[Na]_i$ within a normal physiological range is not necessary to produce a physiological AP waveform, it constrains more tightly the conductance of the $Na^+$-$Ca^{2+}$ exchanger current (compare $I_{NaCa}$ histograms of *Figure 3D* and *Figure 3—figure supplement 1*). This leads in turn to a tighter three-way compensation between the L-type $Ca^{2+}$ current and two dominant repolarizing $K^+$ currents in the mouse, which is less pronounced in a GES search in which $[Na]_i$ is not constrained (*Figure 3—figure supplement 2*) and $I_{Ca,L}$ can be compensated by $I_{NaCa}$ in addition to those $K^+$ currents (four-way compensation). In the present study, we have leveraged the fact that different mice in the same HMDP inbred strain are isogenic to distinguish for the first time intra-heart cell-to-cell variability from inter-subject (inter-strain in the HMPD context) variability. This has allowed us to use several hearts for each strain and perform current measurements in enough cells to statistically distinguish mean conductances of several currents in different strains. The results (*Figure 4A*, *Table 1* and *Figure 4—figure supplement 1*) clearly show that conductances differ between strains. Statistical testing shows that $I_{Ca,L}$ conductance measurements for different strains are very unlikely to belong to the same distribution (as indicated by the very small p-value) and mean conductances can vary by as much as two-hand-a-half fold between pairs of strains (e.g. C57BL/6J and BXA25/PgnJ), far in excess of the typical standard error of the mean. Furthermore, guided by the predictions of our computational GES search, we have also measured conductances of two dominant repolarizing currents in the mouse, $I_{to,f}$ and $I_{Kur}$, to test for the existence of compensation between those currents and $I_{Ca,L}$. The results (*Figure 4A*) show that for eight out of the nine strains in which all three currents were measured, the currents accurately compensate each other as predicted by the GES search in which both the CaT and $[Na]_i$ are constrained. Compensation is evidenced by the linear regression fit of $I_{to,f}+I_{Kur}$ versus $I_{Ca,L}$. One outlier strain (BXA12/PgnJ) deviates from this fit but still falls within the larger ensemble of computationally predicted GES without the $[Na]_i$ constraint. Importantly, cells isolated from mouse strains with very different $I_{Ca,L}$ conductance have statistically indistinguishable contractile function (*Figure 4B*). This suggests that compensation between $I_{Ca,L}$ and $K^+$ currents in different HMDP strains is present to maintain $Ca^{2+}$ homeostasis, as assumed in the computational GES search. As a whole, the results clearly support the hypothesis that $Ca^{2+}$ concentration plays a major role in feedback sensing of cellular activity and regulation of ion channel expression. While more strains would need to be studied to more accurately determine the role of the $Na^+$ concentration, measurements reported in *Figure 4A* suggest that it plays at least an auxiliary role in further constraining conductances beyond $Ca^{2+}$ sensing.

One limitation of the present GES search is that it only identifies possible combinations of electrophysiological parameters underlying a normal cardiac electrophysiological phenotype. However, it cannot by itself predict which GES among those found are actually represented, even approximately, in a genetically diverse population. Another limitation is that dominant $K^+$ currents and AP features are markedly different in mouse than human. While there is no analog of the HMDP for human, or other species (such as rabbit, dog, or pig) with AP features similar to human, it might be possible to extend the present study using renewable cardiomyocytes (CMs) derived from induced pluripotent stem cells (iPSC) as an alternative to the HMDP. However, iPSC-CMs and adult myocytes isolated from intact hearts exhibit quantitative differences in their responses to ionic current perturbations (*Gong and Sobie, 2018*). Therefore, it is unclear whether the variation of ionic conductances in iPSC-CMs of genetically different subjects would be representative of the variation of conductances in intact hearts of the same subjects, which is ultimately relevant for pharmacological treatment of cardiac arrhythmias. In addition, from the results of a recent study of AP variability in cardiomyocytes derived from different human subjects (*Britton et al., 2017*), it is unclear if inter- and intra-subject variability could be statistically distinguished in a large population.

Finally, the correlation between $I_{Ca,L}$ conductance and cardiac hypertrophic response of HMDP strains to sustained $\beta$-adrenergic stimulation (*Figure 6*) also highlights the importance of considering the inherent variability of electrophysiological parameters in a genetically diverse population to interpret the variability of phenotypic response to pharmacological perturbations (*Sarkar et al., 2012*; *Britton et al., 2013*; *Britton et al., 2017*; *Gong and Sobie, 2018*) or stressors (*Rau et al., 2017*; *Santolini et al., 2018*). For example, pharmacological treatment with an anti-arrhythmic L-type calcium channel blocker, or pathologies such as hyperkalemia (elevated potassium level), would be expected to have different effects in different subjects. In the setting of the HMDP, an L-type calcium channel blocker or hyperkalemia would be expected to have a stronger effect on the calcium transient and action potential of mice strains that function under normal conditions with larger $I_{Ca,L}$ and potassium current conductances. Consistent with previous population level studies (*Sarkar et al., 2012*; *Britton et al., 2013*; *Britton et al., 2017*; *Gong and Sobie, 2018*; *Rau et al., 2017*; *Santolini et al., 2018*), taking into account this variability seems ultimately needed to develop personalized therapies for cardiac arrhythmias and heart failure.

**Table 1.** Patch clamp measurements of $I_{Ca,L}$, $I_{to,f}$, and $I_{Kur}$ functional current density.
Mean current density averaged over $n$ cells isolated from multiple hearts for each strain is given together with the standard error.

| Strain | $I_{Ca,L}$ (pA/pF) | n | $I_{Kur}$ (pA/pF) | n | $I_{to,f}$ (pA/pF) | n | $I_{Kss}$ (pA/pF) | n |
|---|---|---|---|---|---|---|---|---|
| A/J | 13.71947 ± 1.23085 | 19 | 11.61804 ± 2.79089 | 13 | 13.03735 ± 1.64818 | 14 | 5.23445 ± 0.48377 | 13 |
| BALB/cByJ | 10.72278 ± 1.3951 | 9 | 11.496 ± 2.03274 | 10 | 7.46938 ± 0.82615 | 9 | 8.197 ± 0.62438 | 10 |
| BTBR T+tf/J | 14.75667 ± 1.14159 | 6 | 16.42364 ± 2.78295 | 11 | 9.271 ± 1.43985 | 10 | 6.093 ± 0.39553 | 9 |
| BXA12/PgnJ | 11.01333 ± 0.98995 | 9 | 15.66429 ± 1.70548 | 8 | 11.89875 ± 2.18027 | 8 | 8.34556 ± 1.30065 | 9 |
| BXA25/PgnJ | 17.57 ± 4.81376 | 5 | 15.8049 ± 1.74509 | 7 | 11.62704 ± 1.59613 | 7 | 7.11986 ± 1.31554 | 6 |
| BXH6/TyJ | 7.32625 ± 0.59327 | 16 | 5.954 ± 0.43731 | 10 | 8.73172 ± 0.67617 | 11 | 7.98545 ± 0.87545 | 11 |
| C57BL/6J | 7.3925 ± 0.93181 | 10 | 7.82727 ± 1.45134 | 11 | 7.594 ± 0.92097 | 10 | 9.16273 ± 1.62148 | 11 |
| CXB1/ByJ | 11.93909 ± 0.81022 | 11 | 11.69556 ± 1.42095 | 9 | 8.73883 ± 1.11087 | 10 | 6.089 ± 0.5724 | 10 |
| CXB11/HiAJ | 7.63625 ± 0.89344 | 8 | 8.90316 ± 2.49755 | 6 | 8.42537 ± 1.24635 | 7 | 6.8481 ± 1.3638 | 7 |
| AXB8/PgnJ | 7.355 ± 1.09612 | 6 | - | | - | | - | |
| BXA14/PgnJ | 11.28091 ± 1.00796 | 15 | - | | - | | - | |
| BXA4/PgnJ | 11.77615 ± 1.19476 | 13 | - | | - | | - | |
| BXD34/TyJ | 11.12556 ± 1.13741 | 9 | - | | - | | - | |
| CBA/J | 7.956 ± 0.81269 | 10 | - | | - | | - | |
| CXB7/ByJ | 9.05386 ± 0.6087 | 16 | - | | - | | - | |
| SJL/J | 11.2745 ± 0.99708 | 20 | - | | - | | - | |

DOI: https://doi.org/10.7554/eLife.36717.010

## Materials and methods

### Overview of the HMDP

The hybrid mouse diversity panel (HMDP) consists of a population of over 100 inbred mouse strains selected for usage in systematic genetic analyses of complex traits (*Ghazalpour et al., 2012*). The main goals in selecting the strains were to (i) increaseresolution of genetic mapping, (ii) have a renewable resource that is available to all investigators world-wide, and (iii) provide a shared data repository (https://systems.genetics.ucla.edu/about/hmdp2) that would allow the integration of data across multiple scales, including genomic, transcriptomic, metabolomic, proteomic, and clinical phenotypes.

### Electrophysiological and contraction measurements

#### Cell isolation

Ventricular myocytes were enzymatically isolated from the hearts of adult female mice (8- to 12-week-old) using a procedure previously developed and utilized to isolate rabbit cardiomyocytes by *Yang et al. (2008)*. Briefly, hearts were removed from mice anesthetized with intravenous pentobarbital and perfused retrogradely at 37°C in Langendorff fashion with nominally $Ca^{2+}$-free Tyrode's buffer containing 1.2 mg/ml collagenase type II (catalog number 4176; Worthington) and 0.12 mg/ml protease type XIV (catalog number P5147; Sigma) for 10 – 17 min. After washing out the enzyme solution, the ventricles were cut from the atria and aorta and transferred to a separate glass dish containing Tyrode's solution. Cells were isolated by gentle mechanical dissociation, stored at room temperature, and used within 5 hr. This procedure typically yielded 30 – 50% of rod-shaped and $Ca^{2+}$-tolerant myocytes.

#### Patch clamping

Isolated ventricular myocytes were patch clamped in the whole cell ruptured patch configuration using borosilicate glass pipettes (1−3-megaohm tip resistance). Myocytes were superfused at 34−36°C with Tyrode's solution modified accordingly. Currents were measured under voltage clamp conditions, using an Axopach 200B amplifier with a Digidata 1440A interface (Axon Instruments, Union City, CA). Data were acquired and analyzed using pClamp (Axon instruments) and Origin (Origin).

#### L-type calcium current measurements

For characterization of $I_{Ca}$ properties, the pipette solution, designed to eliminate $K^+$ and $Cl^-$ currents, contained (in mM) 100 CsMeS, 30 CsCl, 5 MgATP, five phophocreatine di(tris), 5 N-2-hydroxyethylpi-perazine-N'−2-ethanesulfonic acid (HEPES), 5 NaCl and 0.1 1,2-Bis(2-Aminophenoxy)ethane-N,N,N',N'-tetraacetic acid (BAPTA) (pH adjusted with HEPES to 7.1 – 7.2). The superfusate, designed to eliminate $K^+$ currents, contained (in mM), 136 NaCl, 5.4 CsCl, 1 MgC12, 0.33 NaH2PO4, 10 glucose, 5 HEPES, and 1.2 CaC12 (pH adjusted with Trizma base to 7.4). Voltage-clamp protocols to assess activation were as previously described by *Delbridge et al. (1997)*. In all experiments, peak $I_{Ca}$ current was recorded after a voltage step from −50 to 0 mV.

#### Potassium current measurements

For characterization of $K^+$ current properties, the pipette solution, designed to eliminate $Ca^{2+}$ currents, contained (in mM) 130 KCl, 5 MgATP, five phophocreatine di(tris), 5 HEPES, 5 NaCl, and 10 BAPTA (pH adjusted with HEPES to 7.1 – 7.2).The superfusate, designed to eliminate $Ca^{2+}$ currents, contained (in mM) 136 NaCl, 5.4 KCl, 1 MgC12, 0.33 NaH2PO4, 10 glucose, 5 HEPES, and 0.2 CdCl2 (pH adjusted with Trizma base to 7.4). To access the activation of $K^+$ current components: $I_{Kur}$, $I_{to,f}$ and $I_{Kss}$, we adopted a voltage protocol, similar to the one reported by *Zhou et al. (1998)*, in combination with the usage of 4-aminopyridine (4-AP) that has the following pharmacological properties: (1) $I_{Kur}$ is markedly blocked by 4-AP at submillimolar concentration (e.g. 0.1 mM); (2) a higher concentration (i.e., >1 mM) blocks $I_{to,f}$ effectively; and (3) $I_{Kss}$ is 4-AP resistant. Using this procedure, peak $I_{Kur}$, $I_{to,f}$ and $I_{Kss}$ currents could be deduced from three current measurements without 4-AP and with 0.1 mM and 2 mM 4-AP after a voltage step from −50 to 0 mV.

## Results of patch clamp measurements

Results of patch clamp measurements for all strains are summarized in *Table 1* and *Figure 4—figure supplement 1*. Both the L-type $Ca^{2+}$ current and two $K^+$ currents ($I_{to,f}$ and $I_{Kur}$) were measured in nine strains and the L-type $Ca^{2+}$ current alone was measured in 16 strains.

## Contraction analysis

The length of ventricular myocytes was measured and analyzed using the method described by *Sdek et al. (2011)*. Briefly, myocytes were imaged during pacing using a high-speed charge-coupled device-based camera ($128 \times 128$ pixels; Cascade 128+; Photometrics) at 290 frames/s. The acquired video image data were then processed using Imaging Workbench software (version 6.0; INDEC Bio-Systems). Myocyte length was measured and analyzed using ImageJ software. Cell shortening was calculated from the ratio of peak systolic length to resting diastolic length averaged over 10 contractions evoked by the stimulus train. Results of cell shortening measurements for six strains paced at 4 Hz are summarized in *Table 2*.

## Heart extraction and mass measurement for cardiac hypertrophic response

At sacrifice, hearts were excised, drained of excess blood and weighed. Each chamber of the heart (LV with inter-ventricular septum, RV-free wall, RA and LA) was isolated and subsequently weighed. Cardiac hypertrophy was calculated as the increase in total heart weight after isoproterenol (ISO) treatment compared to control animals (see *Table 3*). As described prevoiusly (*Wang et al., 2016*), the ISO treatment consisted of 30 mg per kg body weight per day of Isoproterenol (ISO) administered for 21 days in 8- to 10-week-old female mice using ALZET osmotic mini-pumps, which were surgically implanted intraperitoneally. The average number of control hearts per strain was 2.75. The average number of treated hearts per strain was 3.5. The exact number of hearts per strain can be found in Table S1 of *Santolini et al. (2018)*.

## Mathematical model of mouse ventricular myocytes

We have developed a novel mathematical model of mouse ventricular myocytes that combines elements of previously published ventricular mycoyte models (*Shiferaw et al., 2003*; *Shannon et al., 2004*; *Bondarenko et al., 2004*; *Mahajan et al., 2008*). For this purpose, we kept the mathematical formulation of intracellular calcium cycling of the Mahajan model developed by *Shiferaw et al. (2003)*, which physiologically incorporates graded release by linking the $Ca^{2+}$ spark recruitment rate to $I_{Ca,L}$ current magnitude, and replaced several sarcolemmal currents by those formulated by *Bondarenko et al., 2004* for mouse ventricular mycoytes. This model allowed us to explore efficiently the space of good enough solutions that we can compare to experimentally measured variability found in the hybrid mouse diversity panel (HMDP). The *Bondarenko et al., 2004* mouse model has sarcolemmal currents fitted to detailed experimental measurements of sarcolemmal currents in mouse ventricular myocytes. The *Mahajan et al. (2008)* model is a rabbit model that integrates a Markov model of $I_{Ca,L}$ together with the *Shannon et al. (2004)* formulation of other sarcolemmal currents and the *Shiferaw et al. (2003)* model of calcium cycling and SR calcium release. The *Shiferaw et al. (2003)* model represents the release of calcium from the SR as a sum of individual spark events, which reproduces important observed instabilities such as $Ca^{2+}$ transient

**Table 2.** Cell Shortening at 4 Hz pacing.

| Strain | ∆L/L (4 Hz) | n |
|---|---|---|
| BXA12/PgnJ | $0.0835 \pm 0.0184$ | 5 |
| BXA14/PgnJ | $0.1182 \pm 0.0108$ | 14 |
| BTBR T+tf/J | $0.0978 \pm 0.0095$ | 7 |
| BALB/cByJ | $0.105 \pm 0.0158$ | 5 |
| C57BL/6J | $0.0989 \pm 0.016$ | 6 |
| BXA25/PgnJ | $0.0838 \pm 0.0112$ | 5 |

DOI: https://doi.org/10.7554/eLife.36717.017

alternans. Even though the *Shiferaw et al. (2003)* model of calcium cycling model was developed for rabbit mycoytes, it can in principle describe Ca$^{2+}$ cycling in other species including mouse. Therefore, we constructed a mouse model starting with the *Mahajan et al. (2008)* model, which incorporates the *Shiferaw et al. (2003)* model of calcium cycling, and using the formulation of *Shannon et al. (2004)* for I$_{Ca,L}$ and *Bondarenko et al., 2004* for other sarcolemmal currents fitted to mouse data. We did not use the *Bondarenko et al., 2004* detailed Markov formulations of the gating of I$_{Na}$ and I$_{Ca,L}$ channels with fast transition rates that were computationally prohibitive for the number of simulations we performed in this study. However, we checked that our combined model reproduces well the mouse electrophysiological phenotype of the *Bondarenko et al., 2004* model while being computationally efficient and incorporating a realistic description of Ca$^{2+}$ cycling. We verified that the *Shiferaw et al. (2003)* model of Ca$^{2+}$ cycling integrated with the *Shannon et al. (2004)* model of I$_{Ca,L}$ produced a normal bell-shaped SR release as a function of step-voltage. We also verified that the force-frequency relationship produced by the model has the correct negative staircase observed experimentally. The equations for the model are described below. The reference values of the six parameters varied in this study are given in *Table 4* and produce baseline AP and CaT morphologies consistent with the experimental measurements in *Bondarenko et al., 2004*. *Table 5* lists all other parameters used that were kept fixed.

## Equations for Ca$^{2+}$ cycling

We use the model for Ca$^{2+}$ cycling developed by *Shiferaw et al. (2003)* and subsequently implemented in *Mahajan et al. (2008)*. The equations for Ca$^{2+}$ cycling are

$$\frac{dc_s}{dt} = \beta_s \left[ \frac{v_i}{v_s}(J_{rel} - J_d + J_{Ca} + J_{NaCa}) - J_{trpn}^s \right],$$ (2)

$$\frac{dc_i}{dt} = \beta_i \left[ J_d - J_{up} + J_{leak} - J_{trpn}^i - J_{PMCA} \right],$$ (3)

$$\frac{dc_j}{dt} = -J_{rel} + J_{up} - J_{leak},$$ (4)

$$\frac{dc_j'}{dt} = \frac{c_j - c_j'}{\tau_a},$$ (5)

**Table 3.** Heart mass before and 3 weeks after Isoproterenol (ISO) injection.

| Strain | Heart mass pre-ISO, $m_{pre}$ (g) | Heart mass post-ISO, $m_{post}$ (g) |
|---|---|---|
| A/J | 0.088666667 | 0.133 |
| AXB8/PgnJ | 0.087 | 0.0992 |
| BALB/cByJ | 0.10105 | 0.1389 |
| BTBR T+tf/J | 0.14162 | 0.223 |
| BXA-12/PgnJ | 0.064 | NA |
| BXA-14/PgnJ | 0.0975 | 0.1248 |
| BXA-4/PgnJ | 0.1031 | 0.14675 |
| BXD-34/TyJ | 0.1215 | NA |
| BXH-6/TyJ | 0.0845 | 0.0878 |
| C57BL/6J | 0.096716667 | 0.1222 |
| CBA/J | 0.095333333 | 0.13 |
| CXB-11/HiAJ | 0.1135 | 0.1255 |
| CXB-7/ByJ | 0.109 | 0.1475 |
| SJL/J | 0.087 | 0.123333333 |

DOI: https://doi.org/10.7554/eLife.36717.015

$$J_d = \frac{c_s - c_i}{\tau_s}, \tag{6}$$

$$\frac{dc_p}{dt} = \tilde{J}_{SR} + \tilde{J}_{Ca} - \frac{(c_p - c_s)}{\tau_d}, \tag{7}$$

where the SR leak flux and RyR release flux are given by

$$J_{leak} = g_l(12.4c_j - c_i), \tag{8}$$

$$\frac{dJ_{rel}}{dt} = N'_s(t)c_j \frac{Q(c'_j)}{c_{sr}} - \frac{J_{rel}}{T}, \tag{9}$$

$$T = \frac{\tau_r}{1 - \tau_r\left(\frac{dc_j}{dt}/c_j\right)} \tag{10}$$

$$Q(c'_j) = \begin{cases} 0 & 0 < c'_j < 50, \\ c'_j - 50 & 50 \leq c'_j \leq c_{sr}, \\ uc'_j + (1-u)c_{sr} - 50 & c'_j > c_{sr}, \end{cases} \tag{11}$$

$$N'_s = -g_{RyR}(V)P_o i_{Ca}, \tag{12}$$

$$g_{RyR}(V) = g_{RyR}\frac{e^{-0.05(V+30)}}{1 + e^{-0.05(V+30)}}, \tag{13}$$

$$\tilde{J}_{SR} = -g_{SR}(V)Q(c'_j)P_o i_{Ca}, \tag{14}$$

$$g_{SR}(V) = 50g_{RyR}(V) \tag{15}$$

## Intracellular Ca²⁺ buffering

Similarly to *Mahajan et al. (2008)*. All buffering parameters are experimentally based and summarized in *Shannon et al. (2004)*. Buffering to SR, calmodulin, membrane, and sarcolemma binding sites are modeled using the instantaneous buffering approximation given by

$$\beta_i = \left(1 + \frac{B_{SR}K_{SR}}{(c_i + K_{SR})^2} + \frac{B_{cd}K_{cd}}{(c_i + K_{cd})^2} + \frac{B_{mem}K_{mem}}{(c_i + K_{mem})^2} + \frac{B_{sar}K_{sar}}{(c_i + K_{sar})^2}\right)^{-1}, \tag{16}$$

**Table 4.** Reference values of ionic current parameters varied in the GES search.

| Parameter | Definition | Reference value | Reference source |
|-----------|------------|-----------------|------------------|
| $g_{Ca}$ | Ca²⁺ current flux | 333.32 mmol/(Cm C) | Measured |
| $v_{up}$ | Peak uptake rate | 1.17 $\mu$M/ms | Chosen[*] |
| $g_{NaCa}$ | Peak NaCa rate | 36.6 $\mu$M/s | *Bondarenko et al., 2004* |
| $g_{RyR}$ | Release current strength | 12.9 sparks cm²/mA | *Mahajan et al. (2008)* |
| $g_{to,f}$ | $I_{to,f}$ peak conductance | 0.16 A/F | Measured |
| $g_{Kur}$ | $I_{Kur}$ peak conductance | 0.144 A/F | Measured |

[*]$v_{up}$ was chosen such that the reference Ca²⁺ transient amplitude was normal.

DOI: https://doi.org/10.7554/eLife.36717.018

**Table 5.** Mouse ventricular myocyte model parameters.

| Parameter | Definition | Value |
|---|---|---|
| **Physical constants and ionic concentrations** | | |
| $C_m$ | Cell capacitance | $3.1 \times 10^{-4} \mu F$ |
| $v_i$ | Cell volume | $2.58 \times 10^{-5} \mu l$ |
| $v_s$ | Submembrane volume | $0.02\ v_i$ |
| F | Faraday Constant | 96.485 C/mmol |
| R | Universal gas constant | 8.314 J mol$^{-1}$ K$^{-1}$ |
| T | Temperature | 298 K |
| $[Na]_o$ | External Na$^+$ concentration | 140 mM |
| $[K]_i$ | Internal K$^+$ concentration | 143.5 mM |
| $[K]_o$ | External K$^+$ concentration | 5.4 mM |
| $[Ca^{2+}]_o$ | External Ca$^{2+}$ concentration | 1.8 mM |
| **Cytosolic buffering parameters** | | |
| $B_T$ | Troponin C concentration | 70 $\mu$mol/l cyt |
| $k_{on}^T$ | on rate for Troponin C binding | 0.0327 ($\mu$M ms)$^{-1}$ |
| $k_{off}^T$ | off rate for Troponin C binding | 0.0196 (ms)$^{-1}$ |
| $B_{SR}$ | SR binding site concentration | 47 $\mu$mol/l cyt |
| $K_{SR}$ | SR binding site disassociation constant | 0.6 $\mu$M |
| $B_{Cd}$ | Calmodulin binding site concentration | 24 $\mu$mol/l cyt |
| $K_{Cd}$ | Calmodulin binding site disassociation constant | 7 $\mu$M |
| $B_{mem}$ | Membrane binding site concentration | 15 $\mu$mol/l cyt |
| $K_{mem}$ | Membrane binding site disassociation constant | 0.3 $\mu$M |
| $B_{sar}$ | Sarcolemma binding site concentration | 42 $\mu$mol/l cyt |
| $K_{sar}$ | Sarcolemma binding site disassociation constant | 13 $\mu$M |
| **SR release parameters** | | |
| $\tau_r$ | Spark lifetime | 10 ms |
| $\tau_a$ | NSR-JSR diffusion time | 20 ms |
| u | Release slope | 4 ms$^{-1}$ |
| $c_{sr}$ | Release slope threshold | 90 $\mu$M / l cytosol |
| $\tau_d$ | $c_p$ - $c_s$ diffusion time | 0.50 ms* |

*Table 5 continued on next page*

*Table 5 continued*

| Parameter | Definition | Value |
|---|---|---|
| $\tau_s$ | $c_s$ - $c_i$ diffusion time | 0.75 ms |
| Exchanger, uptake, and SR leak parameters | | |
| $c_{up}$ | Uptake threshold | 0.5 $\mu$M |
| $k_{sat}$ | NaCa saturation threshold | 0.1 |
| $\xi$ | NaCa energy barrier position | 0.35 |
| $K_{m,Nai}$ | Ion mobility constant | 21 mM |
| $K_{m,Nao}$ | Ion mobility constant | 87.5 mM |
| $K_{m,Cao}$ | Ion mobility constant | 1380 $\mu$M |
| $g_l$ | Leak current conductance | $1.74 \times 10^{-5} ms^{-1}$ |
| Ionic current parameters | | |
| $g_{Na}$ | $Na^+$ current conductance | 13 mS/$\mu$F |
| $g_{Na,b}$ | $Na^+$ background current conductance | 0.0026 mS/$\mu$F |
| $g_{Ca,b}$ | $Ca^{2+}$ background current conductance | 0.000367 mS/$\mu$F |
| $\bar{g}_{Ca}$ | Strength of local LCC calcium flux | 9000 mM/(cm C) |
| $g_{K1}$ | $I_{K1}$ conductance | 0.2938 mS/$\mu$F |
| $g_{NaK}$ | $I_{NaK}$ conductance | 1.716 mS/$\mu$F |
| $g_{Kss}$ | $I_{Kss}$ conductance | 0.025 mS/$\mu$F |
| $g_{to,s}$ | $I_{to,s}$ conductance | 0 mS/$\mu$F |
| $J_{PMCA,max}$ | Maximal $J_{PMCA}$ flux | one pA/pF |
| $K_{PMCA}$ | Saturation constant for $Ca^{2+}$ current | 0.5 $\mu$M |
| $P_{Ca}$ | Constant | 0.00054 cm/s |
| $P_{o,max}$ | Constant | 0.083 |

[*]We have reduced this value from the original value of *Mahajan et al. (2008)* so that the $Ca^{2+}$ transient increases when SERCA uptake rate is increased.
DOI: https://doi.org/10.7554/eLife.36717.019

$$\beta_s = \left(1 + \frac{B_{SR}K_{SR}}{(c_s + K_{SR})^2} + \frac{B_{cd}K_{cd}}{(c_s + K_{cd})^2} + \frac{B_{mem}K_{mem}}{(c_s + K_{mem})^2} + \frac{B_{sar}K_{sar}}{(c_s + K_{sar})^2}\right)^{-1} \tag{17}$$

Buffering to Troponin C is given by

$$\frac{d[CaT]_i}{dt} = J^i_{trpn} = k^T_{on}c_i\left(B_T - [CaT]_i\right) - k^T_{off}[CaT]_i, \tag{18}$$

$$\frac{d[CaT]_s}{dt} = J^s_{trpn} = k^T_{on}c_s\left(B_T - [CaT]_s\right) - k^T_{off}[CaT]_s \tag{19}$$

## The SERCA uptake pump
Similarly to *Shiferaw et al. (2003)*.

$$J_{up} = \frac{v_{up} c_i^2}{c_i^2 + c_{up}^2} \tag{20}$$

## Na$^+$ dynamics

Intracellular Na$^+$ dynamics are given by

$$\frac{d[Na^+]_i}{dt} = 100 \frac{C_m}{F v_i} \left( I_{Na} + 3I_{NaCa} + 3I_{NaK} + I_{Na,b} \right) \tag{21}$$

In order to reduce computation time, we have sped up the rate at which the system reaches steady-state by increasing $d[Na^+]_i/dt$ by a factor of 100. This will make sodium converge to steady-state on a time-scale fast enough to perform the number of simulations necessary for this study. Once the cell reaches steady-state, $d[Na^+]_i/dt$ is zero, so this modification will not affect the sodium dynamics at steady-state. By doing this, we can save up to 90% of calculation before the system reaches steady-state.

## Ionic currents

The rate of change of the membrane voltage V is described by the equation

$$\frac{dV}{dt} = -\left( I_{stim} + I_{Ca,L} + I_{PMCA} + I_{NaCa} + I_{Na} + I_{to,f} + I_{to,s} + I_{Kur} + I_{Kss} + I_{K1} + I_{NaK} + I_{Ca,b} + I_{Na,b} \right) \tag{22}$$

$$I_{Ca,L} = \frac{-2F v_i}{C_m} J_{Ca} \tag{23}$$

$$I_{NaCa} = \frac{F v_i}{C_m} J_{NaCa} \tag{24}$$

$$I_{PMCA} = \frac{-2F v_i}{C_m} J_{PMCA} \tag{25}$$

where $I_{stim}$ is the external stimulus current driving the cell.

## The L-type Ca current (I$_{Ca,L}$)

Similarly to the *Shannon et al. (2004)* model,

$$J_{Ca} = g_{Ca} P_o i_{Ca} \tag{26}$$

$$\tilde{J}_{Ca} = -\bar{g}_{Ca} P_o i_{Ca} \tag{27}$$

$$i_{Ca} = \frac{4 P_{Ca} V F^2}{RT} \frac{c_s e^{2VF/RT} - 0.341 [Ca^{2+}]_o}{e^{2VF/RT} - 1} \tag{28}$$

$$P_o = P_{o,max} \times d \times f \times f_{Ca} \tag{29}$$

$$\frac{df_{Ca}}{dt} = 0.12 \left( 1 - f_{Ca} \right) - \frac{1.4025}{1 + \left( 30/c_p \right)^4} f_{Ca} \tag{30}$$

$$d_\infty = \frac{1}{1 + e^{-(V+4.6)/6.3}} \tag{31}$$

$$\tau_d = d_\infty \frac{1 - e^{-(V+4.6)/6.3}}{0.035 \left( V + 4.6 \right)} \tag{32}$$

$$f_\infty = 1 - \frac{1}{1 + e^{-(V+22.8)/6.1}} \tag{33}$$

$$\tau_f = \frac{1}{0.02 - 0.007e^{-(0.0337(V+10.5))^2}} \tag{34}$$

$$\frac{dd}{dt} = \frac{d_\infty - d}{\tau_d} \tag{35}$$

$$\frac{df}{dt} = \frac{f_\infty - f}{\tau_f} \tag{36}$$

## Calcium background leak ($I_{Ca,b}$)

$$I_{Ca,b} = g_{Ca,b}(V - E_{Ca}) \tag{37}$$

$$E_{Ca} = \frac{RT}{2F}\log\left(\frac{[Ca^{2+}]_o}{c_i}\right) \tag{38}$$

## The sarcolemmal Ca$^{2+}$ ATPase ($I_{PMCA}$)

The sarcolemmal Ca$^{2+}$ pump ($I_{PMCA}$) provides another mechanism, in addition to the exchanger ($I_{NaCa}$), for the extrusion of Ca$^{2+}$ ions out of the cell. This pump is not included in *Mahajan et al. (2008)*. We added this current using the formula used by *Bondarenko et al., 2004*.

$$J_{PMCA} = J_{PMCA,max}\frac{c_i^2}{K_{PMCA}^2 + c_i^2} \tag{39}$$

## The Na$^+$-Ca$^{2+}$ exchange flux (NaCa)

Similarly to the *Bondarenko et al., 2004* model,

$$J_{NaCa} = g_{NaCa}\frac{e^{\xi VF/RT}[Na^+]_i^3[Ca^{2+}]_o - e^{(\xi-1)VF/RT}[Na^+]_o^3 c_i}{\left(1 + k_{sat}e^{(\xi-1)VF/RT}\right)\left(K_{m,Nao}^3 + [Na^+]_o^3\right)\left(K_{m,Cao} + [Ca^{2+}]_o\right)} \tag{40}$$

## The fast sodium current ($I_{Na}$)

Similarly to the *Shannon et al. (2004)* model,

$$I_{Na} = g_{Na}m^3 hj(V - E_{Na}) \tag{41}$$

$$\frac{dh}{dt} = \alpha_h(1-h) - \beta_h h \tag{42}$$

$$\frac{dj}{dt} = \alpha_j(1-j) - \beta_j j \tag{43}$$

$$\frac{dm}{dt} = \alpha_m(1-m) - \beta_m m \tag{44}$$

$$\alpha_m = 0.32\frac{V + 47.13}{1 - e^{-0.1(V+47.13)}} \tag{45}$$

$$\beta_m = 0.08e^{-V/11} \tag{46}$$

For $V \geq -40\text{mV}$,

$$\alpha_h = 0 \tag{47}$$

$$\alpha_j = 0 \tag{48}$$

$$\beta_h = \frac{1}{0.13(1 + e^{(V+10.66)/-11.1})} \tag{49}$$

$$\beta_j = 0.3 \frac{e^{-2.535 \times 10^{-7}V}}{1 + e^{-0.1(V+32)}} \tag{50}$$

For $V \leq -40$mV,

$$\alpha_h = 0.135 e^{(V+80)/-6.8} \tag{51}$$

$$\beta_h = 3.56 e^{0.079V} + 3.1 \times 10^5 e^{0.35V} \tag{52}$$

$$\alpha_j = \frac{\left(-1.2714 \times 10^5 e^{0.2444V} - 3.474 \times 10^{-5} e^{-0.04391V}\right) \times (V + 37.78)}{1 + e^{0.311(V+79.23)}} \tag{53}$$

$$\beta_j = \frac{0.1212 e^{-0.01052V}}{1 + e^{-0.1378(V+40.14)}} \tag{54}$$

## Sodium background leak ($I_{Na,b}$)

$$I_{Na,b} = g_{Na,b}(V - E_{Na}) \tag{55}$$

$$E_{Na} = \frac{RT}{F} \log\left(\frac{[\text{Na}^+]_o}{[\text{Na}^+]_i}\right) \tag{56}$$

## Inward rectifier $K^+$ current ($I_{K1}$)

Similarly to the *Bondarenko et al., 2004* model,

$$I_{K1} = g_{K1} \frac{[K^+]_o}{[K^+]_o + 0.21} \left(\frac{V - E_K}{1 + e^{0.0896(V-E_K)}}\right) \tag{57}$$

$$E_K = \frac{RT}{F} \log\left(\frac{[\text{K}^+]_o}{[\text{K}^+]_i}\right) \tag{58}$$

## The fast component of the transient outward $K^+$ current ($I_{to,f}$)

This current is modified from the formulation of *Bondarenko et al., 2004* as:

$$I_{to,f} = g_{to,f} a_{to,f}^3 i_{to,f}(V - E_K) \tag{59}$$

$$\frac{da_{to,f}}{dt} = \frac{a_\infty - a_{to,f}}{\tau_a} \tag{60}$$

$$\frac{di_{to,f}}{dt} = \frac{i_\infty - i_{to,f}}{\tau_i} \tag{61}$$

$$\alpha_a = 0.18264 e^{0.03577(V+45)} \tag{62}$$

$$\beta_a = 0.3956e^{0.06237(V+45)} \tag{63}$$

$$\alpha_i = \frac{0.00152e^{-(V+13.5)/7.0}}{0.067083e^{-(V+33.5)/7.0} + 1} \tag{64}$$

$$\beta_i = \frac{0.0095e^{(V+33.5)/7.0}}{0.051335e^{(V+33.5)/7.0} + 1} \tag{65}$$

$$\tau_a = \frac{1}{\alpha_a + \beta_a} \tag{66}$$

$$\tau_i = \frac{1}{\alpha_i + \beta_i} \tag{67}$$

$$a_\infty = \frac{\alpha_a}{\alpha_a + \beta_a} \tag{68}$$

$$i_\infty = (1-r)\frac{\alpha_i}{\alpha_i + \beta_i} + r \tag{69}$$

where $r = 0.37$ accounts for the presence of a persistent outward potassium current in patch clamp measurements of $I_{to,f}$. We have increased the rates of the inactivation gate ($\alpha_i$ and $\beta_i$) from the original formulation to match experimental measurements of $I_{to,f}$ inactivation rate under voltage clamp.

## The slow component of the transient outward K$^+$ current ($I_{to,s}$)

Similarly to the *Bondarenko et al., 2004* model,

$$I_{to,s} = g_{to,s}a_{to,s}i_{to,s}(V - E_K) \tag{70}$$

$$\frac{da_{to,s}}{dt} = \frac{a_{ss} - a_{to,s}}{\tau_{ta,s}} \tag{71}$$

$$\frac{di_{to,s}}{dt} = \frac{i_{ss} - i_{to,s}}{\tau_{ti,s}} \tag{72}$$

$$a_{ss} = \frac{1}{1 + e^{-(V+22.5)/7.7}} \tag{73}$$

$$i_{ss} = \frac{1}{1 + e^{(V+45.2)/5.7}} \tag{74}$$

$$\tau_{ta,s} = 0.493e^{-0.0629V} + 2.058 \tag{75}$$

$$\tau_{ti,s} = 270.0 + \frac{1050}{1 + e^{(V+45.2)/5.7}} \tag{76}$$

## The ultra-rapidly activating component of the delayed rectifier K$^+$ current ($I_{Kur}/I_{K,slow}$)

Similarly to the *Bondarenko et al., 2004* model,

$$I_{Kur} = g_{Kur}a_{ur}i_{ur}(V - E_K) \tag{77}$$

$$\frac{da_{ur}}{dt} = \frac{a_{ss} - a_{ur}}{\tau_{aur}} \tag{78}$$

$$\frac{di_{ur}}{dt} = \frac{i_{ss} - i_{ur}}{\tau_{iur}} \tag{79}$$

$$\tau_{aur} = 0.493 e^{-0.0629V} + 2.058 \tag{80}$$

$$\tau_{iur} = 1200 - \frac{170}{1 + e^{(V+45.2)/5.7}} \tag{81}$$

We have reduced the timescale of inactivation ($\tau_{iur}$) from the original formulation to match experimental measurements of $I_{Kur}$ inactivation rate under voltage clamp.

## The non-inactivating steady-state K$^+$ current ($I_{Kss}$)

Similarly to the *Bondarenko et al., 2004* model,

$$I_{Kss} = g_{Kss} a_{Kss} (V - E_K) \tag{82}$$

$$\frac{da_{Kss}}{dt} = \frac{a_{ss} - a_{Kss}}{\tau_{Kss}} \tag{83}$$

$$\tau_{Kss} = 39.3 e^{-0.0862V} + 13.17 \tag{84}$$

## The Na$^+$-K$^+$ pump current ($I_{NaK}$)

Similarly to the *Bondarenko et al., 2004* model,

$$I_{NaK} = g_{NaK} f_{NaK} \frac{1}{1 + \left(K_{m,Nai}/[\text{Na}^+]_i\right)^{3/2}} \frac{[\text{K}^+]_o}{[\text{K}^+]_o + K_{m,Ko}} \tag{85}$$

$$f_{NaK} = \frac{1}{1 + 0.1245 e^{-0.1 VF/RT} + 0.01548767 \, \sigma \, e^{-VF/RT}} \tag{86}$$

$$\sigma = \frac{1}{7} \left( e^{[\text{Na}^+]_o/67300} - 1 \right) \tag{87}$$

## Effect of $c_s$-$c_i$ diffusion rate on how the Ca$^{2+}$ transient depends on SERCA

The effect of modifications of the SERCA pump on the steady-state Ca$^{2+}$ transient is shown in *Figure 1*. While increasing SERCA peak uptake current has the effect of sequestering Ca$^{2+}$ back into the SR which would reduce the Ca$^{2+}$ transient, the dominant effect is to increase the Ca$^{2+}$ transient due to higher SR Ca$^{2+}$ load at steady-state. This is consistent with experiments showing restoration of Ca$^{2+}$ transient amplitude when SERCA is up regulated (*del Monte et al., 2002*; *del Monte et al., 1999*).

Initial simulations using the original Ca$^{2+}$ cycling parameters of *Mahajan et al. (2008)* showed that the SR Ca$^{2+}$ load decreased as the uptake rate was increased since more Ca$^{2+}$ was extruded from the sub membrane region of the myocyte via NCX before it had sufficient time to diffuse from the sub membrane space (with local Ca$^{2+}$ concentration denoted by $c_s$) into the cytosol compartment (with local Ca$^{2+}$ concentration denoted by $c_i$) to be re-uptaken into the SR. In order to rectify this, we increased the diffusion rate between the sub membrane and cytosolic compartments. We found that when this rate is faster (smaller $\tau_s$), the SR Ca$^{2+}$ load increases with increasing SERCA uptake rate as experimentally observed. For this reason, we use a value of $\tau_s$ = 0.75 ms for all simulations in this study.

## Computational reproduction of patch clamp experiments

In order to compare the GESs found by the computational search to the phenotype variability found in the HMDP, we iterate the model with voltage held constant, reproducing the experimental patch clamp procedure described in the Methods section of the main text. Each model is simulated for 1 s with $V_m$ held at $-50$ mV in order to reach steady-state. $V_m$ is then raised to 0 mV, and peak values $I_{Ca,L}$, $I_{to,f}$ and $I_{Kur}$ are recorded.

## Tissue scale modeling

Tissue scale modeling is performed using a $56 \times 56$ array of myocytes, each with individual values of ionic conductances. Electrotonic coupling is simulated by introducing a diffusive into the $V_m$ evolution equation,

$$\frac{\partial V_m}{\partial t} = -\frac{1}{C_m}(I_{stim} + I_{ion}) + D\nabla^2 V_m \tag{88}$$

The applied stimulus current occurs at a pacing rate of 4 Hz and is applied to each myocyte simultaneously. The diffusive term is applied isotropically, with diffusive co-efficient, $D = 1\,\mathrm{cm}^2/s$. In the discretized diffusion equation,

$$V_{i,j}^{t+1} = V_{i,j}^t + dt[-\frac{1}{C_m}(I_{stim} + I_{ion}) + \frac{D}{\Delta x^2}(\sum V_{i\pm1,j\pm1}^t - 4V_{i,j}^t)], \tag{89}$$

We use a lattice size of $\Delta x = 225\,\mu m$, such that the $56 \times 56$ lattice represents a 1.25 cm $\times$1.25 cm tissue.

## GES search

In this study, we consider variation in six important ionic currents: L-type Ca current ($I_{Ca,L}$), the SR ATPase SERCA, $Na^+$-$Ca^{2+}$ exchange (NaCa), ryanodine receptor (RyR), the transient outward $K^+$ current ($I_{to,f}$), and the ultra-rapidly-activating $K^+$ current ($I_{Kur}$). Those six currents were selected because they are the major currents influencing the CaT. The strength of these ionic currents is determined by their conductance $g_i$. Any given set of parameters ($\mathbf{p} = \{p_1,\ p_2, \cdots, p_n\}$) corresponds to a different candidate myocyte model and produces a different phenotype, which we characterize by quantifiable measurements of its steady-state behaviour (sensors) that is steady state calcium transient amplitude, action potential duration and sarcoplasmic reticulum (SR) $Ca^{2+}$ concentration. When stimulating a model with a given period, these parameters (once the simulation has reached steady state) produce a phenotype which we can compare to the phenotype produced by the standard parameters of our model ($\mathbf{p_{ref}}$).

We define a cost function $E$ that quantifies how much each model's phenotype differs from our reference phenotype as

$$E(\mathbf{p})^2 = \sum_{n=1}^{N}\left(\frac{S_n(\mathbf{p}) - S_n(\mathbf{p_{ref}})}{S_n(\mathbf{p_{ref}})}\right)^2, \tag{90}$$

where the $S_n(\mathbf{p})$s are sensors characterizing the electrophysiological phenotype of the model's output. $E(\mathbf{p_{ref}})$ is zero by definition. The three sensors used in this study are listed in *Table 6*.

For any given set of conductances, a simulation is performed that outputs a value for each sensor. All values are calculated after the simulation is paced with a pacing cycle length PCL = 250 ms for 12.5 s when the system has reached steady state.

We define a *good enough solution* (GES) as a set of conductances with a phenotype such that the value of cost function $E(\mathbf{p})$ is less than a threshold $\epsilon = 0.05$. None of the cost function sensors $S_n$ are based on the membrane potential, and therefore a GES does not necessarily have an action potential shape close to the reference action potential shape. A GES is required to achieve steady-state. Therefore, parameters that produce parameter sets that do not reach a steady state during pacing at constant cycle length, such as those which exhibit calcium transient alternans, are not considered GESs. We additionally reject any set of parameters for which the output steady-state SR $Ca^{2+}$ load is above a threshold $[Ca^{2+}]_{SR} > 130\ \mu M_{Cyt}$, which we consider to be unphysiologically overloaded.

Completing an exhaustive search of the parameter space becomes increasingly computationally intensive as the number of parameters grows. An exhaustive search of a $M$ dimensional space, considering $K$ possible values for each conductance requires $K^M$ evaluations of the cost function. Additionally, as the number of sensors ($N$) that we use to calculate the cost function increases, the fraction of models tested that are good enough solutions ($\Phi$) will decrease. We calculated $E(\mathbf{p})$ for $10^7$ random parameter sets, $\mathbf{p}$, and found only 11 which were GESs ($E(\mathbf{p})<0.05$). It is therefore apparent that an exhaustive search is not efficient for finding GES in high dimensional parameter space and for this reason we use a minimization scheme to find GESs. We start by randomly assigning values to each parameter such that $0<p_i<3p_{i,ref}$ for each $i$ in $\mathbf{p}$, and then minimize them with respect to the cost function, $E(\mathbf{p})$, using the Nelder-Mead simplex algorithm (*Nelder and Mead, 1965*) (also known as the Amoeba algorithm) until $E(\mathbf{p})<\epsilon$. Running this procudure 10,000 times yeilds 7263 GESs, with the remaining trials being rejected either because the minimization algorithm does not converge, the SR load constraint is not satisfied, or the system is found to not be in steady-state (determined by comparing $\Delta[\text{Ca}]_i$ of the 50th and 51 st beat).

## The Nelder-Mead simplex algorithm

The Nelder-Mead algorithm (*Nelder and Mead, 1965*) maintains a non-degenerate simplex at each iteration, a geometric figure in $n$ dimensions of nonzero volume that is the convex hull of $n+1$ vertices, $\vec{x_0}, \vec{x_1}, ..., \vec{x_n}$, and their respective function values. Suppose we start from the vector $\vec{x_0}$, the simplex can be initialized as $\vec{x_i} = \vec{x_0} + \delta \vec{e_i}$, where $\vec{e_i}$ is a unit vector, and where $\delta$ is our guess of the problem's characteristic length scale. In each iteration, new points are computed, along with their function values, to form a new simplex. The algorithm terminates when the function values at the vertices of the simplex satisfy a predetermined condition. One iteration of the Amoeba algorithm consists of the following steps (the standard values for the coefficients are: $\alpha = 1, \beta = 2, \gamma = 0.5, \sigma = 0.5$):

1. Order: order and re-label the $n+1$ vertices as $\vec{x_0}, \vec{x_1}, ..., \vec{x_n}$, such that $F(\vec{x_0}) \leq F(\vec{x_1}) \leq ... \leq F(\vec{x_n})$. Since we want to minimize $F(\vec{x_0})$, we refer to $x_0$ as the best point, to $\vec{x_n}$ as the worst point, and to $\vec{x_{n-1}}$ as the next worst point. Let $\vec{x_c}$ refer to the centroid of the $n$ points in the vertex.
$\vec{x_c} = \sum\limits_{i=0}^{n-1} \vec{x_i}/n.$
2. Reflect: compute the reflected point, $\vec{x_r} = \vec{x_c} + \alpha(\vec{x_c} - \vec{x_n})$. Evaluate $F(\vec{x_r})$. If $F(\vec{x_0}) \leq F(\vec{x_r})<F(\vec{x_{n-1}})$, then obtain a new simplex by replacing the worst point $\vec{x_n}$ with the reflected point $\vec{x_r}$ and go to step 1.
3. Expand: if $F(\vec{x_r})<F(\vec{x_0})$, compute the expanded point, $\vec{x_e} = \vec{x_c} + \beta(\vec{x_r} - \vec{x_c})$. If $F(\vec{x_e})<F(\vec{x_r})$, then obtain a new simplex by replacing the worst point $\vec{x_n}$ with the expanded point $\vec{x_e}$ and go to step 1; otherwise then obtain a new simplex by replacing the worst point $\vec{x_n}$ with the reflected point $\vec{x_r}$ and go to step 1.
4. Contract: At this step, where it is certain that $F(\vec{x_r})>F(\vec{x_{n-1}})$, compute the contracted point $\vec{x_{con}} = \vec{x_c} + \gamma(\vec{x_n} - \vec{x_c})$. If $F(\vec{x_{con}}) \leq F(\vec{x_n})$, obtain a new simplex by replacing the worst point $\vec{x_n}$ with the expansion point $\vec{x_{con}}$ and go to step 1.
5. Shrink: replace all vertices except the best $\vec{x_0}$ with $\vec{x_i} = \vec{x_0} + \sigma(\vec{x_i} - \vec{x_0})$ and go to step 1.

## Two sensor search only constraining the Ca$^{2+}$ transient

The results of the GES search described in the main text were reproduced using only two sensors, constraining $\langle[\text{Ca}]_i\rangle$ and $\Delta[\text{Ca}]_i$ but not constraining $[\text{Na}]_i$. This results in a broader histogram for the $I_{\text{NaCa}}$ conductance (*Figure 3—figure supplement 1* compared to *Figure 3*).

**Table 6.** Simulation outputs corresponding to physiological sensors

| Abbreviation | Description | Reference value |
|---|---|---|
| $\langle[\text{Ca}]_i\rangle$ | Average cytostolic Ca$^{2+}$ over one beat | 0.24 $\mu$M |
| $\Delta[\text{Ca}]_i$ | Ca$^{2+}$ transient amplitude | 0.5 $\mu$M |
| $[\text{Na}]_i$ | Diastolic Na$^+$ | 14 mM |

DOI: https://doi.org/10.7554/eLife.36717.020

## Code availability

The codes used in this work are available at: https://github.com/circs/GES (*Rees, 2018*; copy archived at https://github.com/elifesciences-publications/GES).

## Acknowledgements

This research was supported by NIH/NHLBI grant 5R01HL114437 and by the Laubisch and Kawata Endowments. The authors thank Yibin Wang for valuable input into this study. AK acknowledges stimulating discussions with Eve Marder in the early stage of this work.

## Additional information

### Funding

| Funder | Grant reference number | Author |
| --- | --- | --- |
| National Heart, Lung, and Blood Institute | 5R01HL114437 | Colin M Rees<br>Jun-Hai Yang<br>Marc Santolini<br>Aldons J Lusis<br>James N Weiss<br>Alain Karma |

The funders had no role in study design, data collection and interpretation, or the decision to submit the work for publication.

### Author contributions

Colin M Rees, Software, Formal analysis, Investigation; Jun-Hai Yang, Data curation, Investigation, Writing—review and editing, Made a very substantial contribution to this work, Performed all the patch clamp experiments and contractility measurements, Recorded three major ionic currents and cell shortening for a large number of cells and different mice strains, Provided key data to validate computational modeling predictions, Responsible for curation of the data, Helped review the draft of the paper; Marc Santolini, Data curation, Validation, Writing—review and editing; Aldons J Lusis, Resources, Funding acquisition; James N Weiss, Conceptualization, Supervision, Funding acquisition; Alain Karma, Conceptualization, Supervision, Funding acquisition, Methodology, Writing—original draft

### Author ORCIDs

Marc Santolini (iD) http://orcid.org/0000-0003-1491-0120
Alain Karma (iD) http://orcid.org/0000-0001-7032-9862

### Ethics

Animal experimentation: This study was approved by the UCLA Chancellor's Animal Research Committee (ARC 2003-063-23B) and performed in accordance with the Guide for the Care and Use of Laboratory Animals published by the United States National Institutes of Health (NIH Publication No. 85-23, revised 1996) and with UCLA Policy 990 on the Use of Laboratory Animal Subjects in Research (revised 2010).

### Decision letter and Author response

Decision letter https://doi.org/10.7554/eLife.36717.025
Author response https://doi.org/10.7554/eLife.36717.026

## Additional files

### Supplementary files

• Transparent reporting form
DOI: https://doi.org/10.7554/eLife.36717.021

## Data availability

Gene expression data has been deposited in GEO under accession code GSE48760

The following previously published dataset was used:

| Author(s) | Year | Dataset title | Dataset URL | Database and Identifier |
|---|---|---|---|---|
| Rau CD, Wang J, Wang Y, Lusis AJ | 2013 | Transcriptomes of the hybrid mouse diversity panel subjected to Isoproterenol challenge | https://www.ncbi.nlm.nih.gov/geo/query/acc.cgi?acc=GSE48760 | NCBI Gene Expression Omnibus, GSE48760 |

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
