## [Decision Letter]

Thank you for submitting your article "Variability and compensation of cardiomyocyte ionic conductances at the population level" for consideration by *eLife*. Your article has been reviewed by Richard Aldrich as the Senior Editor, a Reviewing Editor, and three reviewers. The following individuals involved in review of your submission have agreed to reveal their identity: David Christini (Reviewer #1), Blanca Rodriguez (Reviewer #2); Colleen Clancy (Reviewer #3).

The reviewers' comments are appended below. As you can see, the reviewers raise a number of questions and concerns pertaining the methodology used in your study and the interpretation and presentation of your results. Based on this assessment, we cannot accept your manuscript for publication in its present form. Nonetheless, the reviewers and the Reviewing Editor concur that the issues raised can be resolved in a reasonable timeframe, and therefore we would like to encourage to resubmit a revised version of your manuscript that addresses the reviewers' comments, along with a point-by-point response

Reviewer #1:

The authors present an intriguing study that adds new insight to a developing area of research utilizing rich electrophysiological and/or clinical data sets to reparameterize cellular cardiac models.

In this study, good enough solution (GES) parameterizations are first developed in simulations in which calcium dynamic properties were constrained by requiring minimization of an error function of calcium and sodium concentration differences. Such constraints produced model variants that illuminated a very interesting compensatory relationship between inward Ca^2+^ and outward K^+^ currents. These findings were used as evidence in support of the "central hypothesis that parameters are constrained predominantly by features of the CaT". I am confused by the logic flow that led to that hypothesis. One issue that comes to mind is that in real cells, how can the authors know if it is CaT that is controlling (via feedback) the ionic current expression (the suggested mechanism), OR if it is the other way around, e.g., inward Ca^2+^ channel expression governing outward K^+^ channel expression (and/or vice versa, via any number of unknown second-messenger mechanisms), which might result in consistent (constrained) CaT?

In several locations, especially in the background and conclusions, the authors emphasize that by focusing on CaT as the main model constraining variables, this study presents a fundamentally new approach. In my view, the approach used in this study is a new twist, rather than a fundamental advancement. Although I did not reread the papers (cited in the background) that used related approaches, my recollection is that some of them at least refer to the idea that adding calcium measurements to their approaches would add value. I thus see this approach as a straightforward variant of previous approaches.

By focusing so much attention on the newness of the CaT constraint, I think the authors distract from two aspects that are more likely to be of interest to the audience: (1) the compensatory Ca^2+^ / K^+^ findings and (2) the utilization of the multiple species of the Hybrid Mouse Diversity Panel. Both of these are strengths of the paper.

It is not clear to me why the authors choose not to include membrane potential in the error function. As they note in subsection “Computationally determined good enough solutions”, "the AP waveforms exhibit larger variations owing to the fact that the GES search does not involve any voltage sensing." Because of this choice, AP waveforms are often non-physiological (e.g. the huge variation shown in Figure 3B), which is a notable weakness.

Overall, this study is intriguing, notwithstanding the aforementioned issues.

Reviewer #2:

The primary aim of the paper is to evaluate how parameters in a mouse cardiac electrophysiology model would be constrained using calcium transient values rather than action potential values. They show that, in mice, constraining the model search using calcium transient yields a correlation between the sum of two potassium currents and the main calcium current.

The idea is interesting. The paper however is superficial in the results provided and also the discussion. This is primarily related to the fact that a comparison of results with action potential calibration versus calcium calibration is not shown, and the results seem rather species dependent (to mice). Both aspects could be easily explored in more depth. I also have further suggestions to better position this work in the context of previous relevant studies in cardiac electrophysiology.

The paper postulates that "model parameters are predominantly constrained by feedback sensing of Ca^2+^, and potentially other ions (e.g. Na^+^) affecting ion channel regulation".

The most interesting result in the paper is the correlation between the sum of two potassium currents and I_Ca,L_, demonstrated both computationally and experimentally. Even though correlations between 2 and 3 parameters have been previously explored and this could be mentioned in the paper (e.g. Sanchez et al., 2014, Britton et al., 2013, Muszkiewicz et al., 2018), their sum hasn't been considered and this seems important. I have two general comments about this:

1) Only 6 parameters are varied and, of these, only 3 conductances are shown to be correlated. The statement above therefore only applies to a subset of conductances. Moreover, previous work on populations of models has shown that certain conductances are indeed constrained when calibrating only with voltage biomarkers (for example the sodium current by the upstroke velocity). How does this fit in the primary hypothesis of the paper? Would those conductances have voltage sensors? Please add discussion on this.

2) The analysis of the AP phenotypes following the Ca-based calibration would be important (it is in fact mentioned in the Discussion but not shown in the results). A previous study shows that AP calibration leads to variable calcium phenotypes in human atrial cells (Muszkiewicz et al., 2018). Something similar can happen with Ca-based calibration. The analysis would be both feasible and very interesting.

3) What does AP calibration (in addition to Calcium calibration) yield in terms of ionic conductances correlation and constrain? The study does not demonstrate that AP calibration doesn't achieve the same sort of correlation in the ionic conductances than calcium transient calibration. This would be important to demonstrate the main premise of the paper.

4) This paper focuses on mice (rather than human or other species). This should certainly be highlighted in the title and somewhat detracts from the relevance of the study. The repolarization ionic currents, action potential and calcium waveforms are very different in mice and human, particular from the ventricles. This is an aspect that should be analysed and discussed in the paper.

It would indeed be interesting to test whether a similar level of conductances correlation is observed or not when using human models (for atria and ventricles) calibrated with calcium data. How many repolarization currents would need to be considered in the sum to obtain correlation with the calcium current? This is feasible given the availability of human models and calcium data, and the fact that the models have already been used for population studies on different human cell types (ventricular, atrial and iPS, see references mentioned above). A similar comparison was for example done in the study by Devenyi and Sobie, 2016.

5) I am confused by what is shown in Figure 5, and how it contributes to the aim of the paper. It shows that electrotonic coupling reduces APD heterogeneity (as shown previously in the context of alternans Zaniboni et al., 2016 as well as with populations of models for example in Sanchez et al., 2017). The paper states that calcium amplitude heterogeneity is also reduced (subsection “Compensation at the organ scale”). However, I see a very small reduction of calcium amplitude heterogeneity in Figure 5E and I certainly don't see a very strong effect. I would expect however spatial heterogeneity of calcium amplitude to be reduced by calcium flow through gap junctions. This model feature seems crucial in this context. In addition, APD calibration would also achieve the same outputs, wouldn't it?

6) The section on "Cardiac hypertrophic response to a stressor" contains the interesting correlation shown in Figure 6 and then it is mostly speculative. I don't see a clear contribution towards the goals of the paper as set in the introduction.

Reviewer #3:

In paper from Rees and colleagues, the authors present a model to predict variability of ion channel conductances in a population of subjects with variable cardiac gene expression patterns. Unlike previous models that account for variability using features of the action potential, the new model focuses on the calcium transient (CaT).

The authors build on previous work demonstrating that many distinct parameter sets can elicit the same phenomenological behavior. Specifically, the authors point out that most previous works have focused on attributes of transmembrane potential as the important phenomenological behaviors to be replicated by all parameter sets. The authors propose that sensory mechanisms exist for monitoring the calcium transient and intracellular sodium dynamics. The authors vary six ionic conductances and examine what combinations replicate the calcium transient and intracellular sodium dynamics of the baseline model. Through this analysis, they are able to identify parameters that are highly constrained, relatively insensitive, or highly correlated with other parameters. As pointed out by the authors, this novel work using the calcium transient and intracellular sodium dynamics as opposed to transmembrane potential dynamics to constrain the model is important, because when models are fit to replicate physiological phenomena, the final model parameters critically depend on what physiological characteristics are used to constrain the model. To test their model, the authors used experiments using the Hybrid Mouse Diversity Panel.

The authors state at the end of their discussion, that this work has very broad implications in the realm of understanding how drugs will affect cardiac health in a diverse population. Since ion channel expression levels can span a wide range (both among individuals in a population as well as between cells in a single individual), the effect of a drug that binds to a specific ion channel could vary widely between individuals or from cell to cell. Therefore, to accurately characterize the effects of a drug that interacts with cardiac ion channels across an entire population, it is important to fully characterize the parameter space of physiologically allowable ionic conductances. This is an important point and could be emphasized more.

What seems to be missing from the study at present is a description of the fundamental insight that is gained by the Ca-based GES search compared to the V-based GES search. It's hard to imagine that the V-based search wouldn't reveal the relationship between I_to,f_ and I_Kur_ and I_Ca,L_. Although I understand that the Ca signal is (likely) more directly related to ion-channel regulation, there is no particular control mechanism included in the analysis – so what does the current state analysis provide that the V-based doesn't? It would be so useful to show a direct comparison of the Ca- and V-based GES searches. It should also be mentioned that there are a host of regulatory mechanisms not accounted for in either search method.

1) More discussion of the range of APs waveforms observed would be helpful, since that is what has been examined in all other cardiac AP population approaches. Additionally, in the Discussion section ("physiological AP waveform whose duration matches the experimentally observed variation in isolated myocytes even though the voltage signal is not used to constrain model parameters"), doesn't seem to be entirely supported by the data shown. Figure 3B would better support this argument if the full range of simulated AP behaviors were quantified and compared to experimental AP data. It is not clear how the APs shown in Figure 3B (and later in Figure 5) were chosen, or if they represent the full range of behaviors observed with this approach.

2) Similarly, for Figure 5C, it would be clearer if cell-to-cell variability in the simulated APs was quantified and compared that to the experimental range. It seems that the variability in the AP waveforms shown in Figure 5C represent a much wider range of morphologies (in APD_90_, for example), than in the experimental APs shown in Figure 5—figure supplement 1.

3) It is not clear what is defined as the "normal CaT (Calcium Transient)", or how that was decided. The reference values (G_ref_) are defined as the conductance in the baseline model, which produce the normal CaT. Is the normal CaT the same as the reference outputs defined in Table 6? Are these the only two outputs used to define the normal CaT? Also, for the reference CaT outputs (as shown in Table 6), how were these values decided? Include a citation (or justification) for the source of the reference/normal values.

4) In the populations shown in Figure 5, is a similar coupling effect seen for average cytosolic Ca^2+^ over one beat? Does this range of simulated values, for either CaT measure, mimic the experimental range?

5) My understanding is that the point of Figure 5 is that even if the conductances considered here aren't tightly correlated, the Ca2+ transient and especially the AP are constrained due to electrical coupling in tissue. However, what is not clear to me based on the authors' discussion is if electrical coupling is enough to constrain the Ca2+ transient and AP to within a healthy range. Based on looking at the figure, electrical coupling does not constrain the Ca2+ transient to within the 5% error used in the earlier good enough solutions. But how well constrained do the Ca2+ transient and AP really need to be? Also, how would things change if the parameter sets of the cells in the tissue came from the good enough solution?

[Editors' note: further revisions were requested prior to acceptance, as described below.]

Thank you for resubmitting your work entitled "Variability and compensation of cardiomyocyte ionic conductances at the population level" for further consideration at *eLife*. Your revised article has been favorably evaluated by Richard Aldrich (Senior Editor), José D. Faraldo-Gómez as Reviewing Editor, and three reviewers.

The manuscript has been improved but there are some remaining issues that we would like you to address before acceptance, as outlined below. Although your revised manuscript will not be sent to the reviewers, we encourage you to carefully consider their suggestions.

Reviewer #1:

The authors have done a good job of addressing the general concerns that I raised in my original review. Notably, they have sharpened up the presentation of the overall hypothesis of CaT regulation, via blood-pressure feedback, serving as the mechanism governing underlying levels of Ca-cycling proteins and the action potential. As I noted in my original review, the computational findings are supported by analyses of the Hybrid Mouse Diversity Panel, which strengthen the study significantly and set it apart from a purely computational study.

One concern I have relates to a point made in response to my review, as well as the other reviewers (and in the Discussion section of the revised manuscript). The authors provide results and analyses to defend the approach of focusing on calcium for the GES determination, rather than also incorporating the action potential. They argue that the variation in the action potential waveform that occurs using the CaT-based parameterization is within expected physiological variability, especially if we expand our definition of physiological variability to include outlier AP waveforms that are excluded via selection bias by experimentalists who "often" discard sick-looking AP waveforms. My first problem here is that this statement is an over simplification of what good experimentalists do and why. Sure, some may just exclude APs that look wrong, but that is not good accepted practice. Instead, they use specific quantifiable criteria, such as minimum resting membrane potential, which have been demonstrated to be correlated to damaged cells, for exclusion. While such criteria may still be excluding physiological APs, my point is that it is more nuanced than the text implies and, I think, does a disservice to the actual experiments and the reported data. Beyond that issue, within the very same paragraph, the authors note that the variability that occurs in CaT using voltage-based parameterization is "highly variable and often nonphysiological." So why is it that the variability that arises using the approach that fits with the authors hypothesis is physiological, whereas the variability using the approach that does not fit the authors hypothesis is "nonphysiological?" Put another way, when the authors define nonphysiological CaT, are they assuming that experimentalists who recorded and reported CaT have never thrown away "sick looking" calcium transient data, which, if included in the literature, would expand the range of physiological? This point is really just about framing, rather than about the overall justification of the approach, as I do agree with the overall conclusion that these findings support using CaT. But I think that point can be made without the "sick looking" discussion or expanding that discussion with more context.

I agree with the point made by referee #2 that it would be useful for the authors to deposit the code used to generate the results.

Reviewer #2:

The authors have followed most of our suggestions and the paper is stronger. The title remains too generic and would benefit from at least mentioning calcium as the main novelty of the paper (and perhaps mice too). Otherwise it could very well be another review on the topic.

---

## [Author Response]

Reviewer #1:The authors present an intriguing study that adds new insight to a developing area of research utilizing rich electrophysiological and/or clinical data sets to reparameterize cellular cardiac models.In this study, good enough solution (GES) parameterizations are first developed in simulations in which calcium dynamic properties were constrained by requiring minimization of an error function of calcium and sodium concentration differences. Such constraints produced model variants that illuminated a very interesting compensatory relationship between inward Ca^2+^ and outward K^+^ currents. These findings were used as evidence in support of the "central hypothesis that parameters are constrained predominantly by features of the CaT". I am confused by the logic flow that led to that hypothesis. One issue that comes to mind is that in real cells, how can the authors know if it is CaT that is controlling (via feedback) the ionic current expression (the suggested mechanism), OR if it is the other way around, e.g., inward Ca^2+^ channel expression governing outward K^+^ channel expression (and/or vice versa, via any number of unknown second-messenger mechanisms), which might result in consistent (constrained) CaT?

Our hypothesis is that the CaT is critical for generating blood pressure, which is sensed by the carotid baroreceptors and feeds back through the autonomic nervous system to regulate the CaT via controlling levels of Ca-cyling proteins and the AP in a way that preserves blood pressure. This provides a very straightforward physiological mechanism that we show not only constrains the CaT to a physiological waveform, but, as an added and novel bonus, also constrains AP features through the Ca current-K current ratio. We have revised the text in the introduction to more clearly state this hypothesis.

In several locations, especially in the background and conclusions, the authors emphasize that by focusing on CaT as the main model constraining variables, this study presents a fundamentally new approach. In my view, the approach used in this study is a new twist, rather than a fundamental advancement. Although I did not reread the papers (cited in the background) that used related approaches, my recollection is that some of them at least refer to the idea that adding calcium measurements to their approaches would add value. I thus see this approach as a straightforward variant of previous approaches.

We believe that the physiological hypothesis, as articulated above, is the novel insight from this study, rather than the technical approach underlying the GES search, which we acknowledge uses standard methods. We have made this clear in the text.

By focusing so much attention on the newness of the CaT constraint, I think the authors distract from two aspects that are more likely to be of interest to the audience: (1) the compensatory Ca^2+^ / K^+^ findings and (2) the utilization of the multiple species of the Hybrid Mouse Diversity Panel. Both of these are strengths of the paper.It is not clear to me why the authors choose not to include membrane potential in the error function. As they note in subsection “Computationally determined good enough solutions”, "the AP waveforms exhibit larger variations owing to the fact that the GES search does not involve any voltage sensing." Because of this choice, AP waveforms are often non-physiological (e.g. the huge variation shown in Figure 3B), which is a notable weakness.Overall, this study is intriguing, notwithstanding the aforementioned issues.

We have given more emphasis to the compensatory Ca/K findings and the utilization of the Hybrid Mouse Diversity Panel in the introduction. We note that although AP waveforms are uniform in tissue, there is a marked variation in isolated ventricular myocytes. Moreover, experimentalists often exclude atypical “sick-looking” AP waveforms in patch clamp experiments, which are assumed to be “non-physiological” due to cell damage during enzymatic isolation or patching. Thus, AP waveform variation in single myocytes may be underestimated because of this selection bias. In coupled tissue, however, occasional atypical AP waveforms would be voltage-clamped by their “healthy” AP waveform neighbors to produce an overall uniform AP waveform. If we had included voltage sensors together with Ca^2+^ sensors, the AP waveforms would, like the CaT, have been much more uniform. However, we did not feel it necessary, since (1) the physiological basis for voltage sensors is not clear, whereas the physiological basis for the CaT as a surrogate for blood pressure mediated feedback to the heart is well-known; (2) if we had included voltage sensors to constrain AP waveform, its variation could become arbitrarily narrower than the observed AP waveform variation in isolated myocytes. In response to this comment and to the similar first major comment of Reviewer #2, we have discussed these points and added new simulation results in which the Ca^2+^ sensors were substituted for voltage sensors (new Figure 3—figure supplement 3). Although the AP waveform was readily constrained, the CaT was highly variable and often nonphysiological, and the Ca/K current ratio relationship was no longer preserved, since other inward and outward currents (in particular the Na^+^-Ca^2+^ exchanger) could regulate AP duration when the CaT was not constrained.

Reviewer #2:The primary aim of the paper is to evaluate how parameters in a mouse cardiac electrophysiology model would be constrained using calcium transient values rather than action potential values. They show that, in mice, constraining the model search using calcium transient yields a correlation between the sum of two potassium currents and the main calcium current.The idea is interesting. The paper however is superficial in the results provided and also the discussion. This is primarily related to the fact that a comparison of results with action potential calibration versus calcium calibration is not shown, and the results seem rather species dependent (to mice). Both aspects could be easily explored in more depth. I also have further suggestions to better position this work in the context of previous relevant studies in cardiac electrophysiology.

As already mentioned in the last response to reviewer #1, we have added new simulation results voltage sensing alone (new Figure 3—figure supplement 3) that can be readily compared to Ca^2+^ sensing alone. With voltage sensing alone, the AP waveform was readily constrained as expected, but the CaT was highly variable and often nonphysiological. The Ca/K current ratio relationship was no longer preserved, since other inward and outward currents could regulate AP duration when the CaT was not constrained. As noted above, our hypothesis provides a clear physiological link for Ca^2+^ sensing, since the CaT is critical for generating blood pressure, which is sensed by the carotid baroreceptors and feeds back through the autonomic nervous system to regulate the CaT by controlling levels of Ca-cyling proteins and the AP in a way that preserves blood pressure. This provides a very straightforward physiological mechanism that not only constrains the CaT to a physiological waveform, but also, as an added and novel bonus, constrains AP features through the Ca current-K current ratio. It is much less clear, on the other hand, how voltage would be sensed by the heart to provide a feedback mechanism to control both the AP waveform and the CaT. Voltage-sensing alone provided a reliable AP waveform, but a highly unreliable CaT, as now shown in the new results. We have emphasized these points in the revised Discussion section.

The paper postulates that "model parameters are predominantly constrained by feedback sensing of Ca^2+^, and potentially other ions (e.g. Na^+^) affecting ion channel regulation". The most interesting result in the paper is the correlation between the sum of two potassium currents and I_Ca,L_, demonstrated both computationally and experimentally. Even though correlations between 2 and 3 parameters have been previously explored and this could be mentioned in the paper (e.g. Sanchez et al., 2014, Britton et al., 20134, Muszkiewicz et al., 2018), their sum hasn't been considered and this seems important. I have two general comments about this:

We have mentioned the additional references in the revised paper.

1) Only 6 parameters are varied and, of these, only 3 conductances are shown to be correlated. The statement above therefore only applies to a subset of conductances. Moreover, previous work on populations of models has shown that certain conductances are indeed constrained when calibrating only with voltage biomarkers (for example the sodium current by the upstroke velocity). How does this fit in the primary hypothesis of the paper? Would those conductances have voltage sensors? Please add discussion on this.

The 6 parameters that we used play a major role in regulating the CaT, based on our hypothesis that the CaT is a surrogate for blood pressure that feeds back to the heart via the autonomic nervous system. The reviewer makes a good point about the Na current, since the integrated CaT of the ventricles has to be reasonably synchronous to generate a normal blood pressure waveform, requiring the Na current density to be adequate for a normal conduction velocity through the tissue. Thus, it seems reasonable to propose that Na current density (and gap junction coupling) could also be regulated by Ca^2+^ sensing to ensure that conduction velocity is adequate to generate a synchronous blood pressure waveform. We have added these points to the Discussion section.

2) The analysis of the AP phenotypes following the Ca-based calibration would be important (it is in fact mentioned in the Discussion but not shown in the results). A previous study shows that AP calibration leads to variable calcium phenotypes in human atrial cells (Muszkiewicz et al., 2018). Something similar can happen with Ca-based calibration. The analysis would be both feasible and very interesting.

Our results show that CaT calibration results in considerable AP waveform variability (Figure 5C) as expected if the AP waveform is not constrained. While this is consistent with the converse finding by Muszkiewicz et al., that the CaT is highly variable when the AP waveform is constrained, constraining the AP waveform does not produce the correlation between Ca and K currents observed in the HMDP, and is difficult to justify biologically given the absence of voltage sensing mechanism. At the tissue level, AP waveform variability is greatly smoothed by diffusive coupling, and so does not require tight control at the single myocyte level. As discussed above, the single myocyte AP variability is considerable when measured experimentally in patch clamp studies. We have added those points to the Discussion section.

3) What does AP calibration (in addition to Calcium calibration) yield in terms of ionic conductances correlation and constrain? The study does not demonstrate that AP calibration doesn't achieve the same sort of correlation in the ionic conductances than calcium transient calibration. This would be important to demonstrate the main premise of the paper.

As noted above, we have added new simulations directly comparing Ca^2+^ sensing alone with voltage sensing alone (Figure 3—figure supplement 3). With voltage sensing alone, the AP waveform was readily constrained as expected, but the CaT was highly variable and often nonphysiological. The Ca/K current ratio relationship was no longer preserved, since other inward and outward currents could regulate AP duration when the CaT was not constrained.

4) This paper focuses on mice (rather than human or other species). This should certainly be highlighted in the title and somewhat detracts from the relevance of the study. The repolarization ionic currents, action potential and calcium waveforms are very different in mice and human, particular from the ventricles. This is an aspect that should be analysed and discussed in the paper.It would indeed be interesting to test whether a similar level of conductances correlation is observed or not when using human models (for atria and ventricles) calibrated with calcium data. How many repolarization currents would need to be considered in the sum to obtain correlation with the calcium current? This is feasible given the availability of human models and calcium data, and the fact that the models have already been used for population studies on different human cell types (ventricular, atrial and iPS, see references mentioned above). A similar comparison was for example done in the study by Devenyi and Sobie, 2016.

We have performed a similar GES study using a rabbit ventricular myocyte model, in which the AP and EC coupling features are closer to human. Although we obtained similar results, we elected not to include these findings in the current manuscript to avoid making it unduly long, and also because a rabbit analog of the HMDP is not available to validate the computational results at a level comparable to what has been achieved for mice here. However, we plan to submit the rabbit model results in a separate manuscript. The results of this GES search is illustrated above for the reviewers’ benefit. We have also added a paragraph in the discussion to suggest the use of cardiomyocytes (CMs) derived from induced pluripotent stem cells (iPSC) as an alternative to the HMDP to extend the present study to human. However, iPSC-CMs and adult myocytes isolated from intact hearts exhibit quantitative differences in their responses to ionic current perturbations (Gong and Sobie, 2018). Therefore, it is unclear whether the variation of ionic conductances in iPSC-CMs of genetically different subjects, if present and statistically distinguishable between subjects, would be representative of the variation of conductances in intact hearts of the same subjects, which is ultimately relevant for pharmacological treatment of cardiac arrhythmias.

5) I am confused by what is shown in Figure 5, and how it contributes to the aim of the paper. It shows that electrotonic coupling reduces APD heterogeneity (as shown previously in the context of alternans Zaniboni et al., 2016 as well as with populations of models for example in Sanchez et al., 2017). The paper states that calcium amplitude heterogeneity is also reduced (subsection “Compensation at the organ scale”). However, I see a very small reduction of calcium amplitude heterogeneity in Figure 5E and I certainly don't see a very strong effect. I would expect however spatial heterogeneity of calcium amplitude to be reduced by calcium flow through gap junctions. This model feature seems crucial in this context. In addition, APD calibration would also achieve the same outputs, wouldn't it?

We agree that unlike APD heterogeneity, electrotonic coupling had a more modest effect on CaT heterogeneity, although it was still significant. It is possible that the heterogeneity would be further reduced by Ca flow through gap junctions, but we did not simulate this possibility. We doubt that this effect would be very significant since Ca diffusion through gap junctions is orders of magnitude slower than voltage diffusion.

6) The section on "Cardiac hypertrophic response to a stressor" contains the interesting correlation shown in Figure 6 and then it is mostly speculative. I don't see a clear contribution towards the goals of the paper as set in the introduction.

We acknowledge that this section is very speculative. However, we prefer to keep it as an example of how the approach outlined in the paper could be applied to obtain insights (or stimulate future avenues of investigation) relevant to disease conditions.

Reviewer #3:In paper from Rees and colleagues, the authors present a model to predict variability of ion channel conductances in a population of subjects with variable cardiac gene expression patterns. Unlike previous models that account for variability using features of the action potential, the new model focuses on the calcium transient (CaT).The authors build on previous work demonstrating that many distinct parameter sets can elicit the same phenomenological behavior. Specifically, the authors point out that most previous works have focused on attributes of transmembrane potential as the important phenomenological behaviors to be replicated by all parameter sets. The authors propose that sensory mechanisms exist for monitoring the calcium transient and intracellular sodium dynamics. The authors vary six ionic conductances and examine what combinations replicate the calcium transient and intracellular sodium dynamics of the baseline model. Through this analysis, they are able to identify parameters that are highly constrained, relatively insensitive, or highly correlated with other parameters. As pointed out by the authors, this novel work using the calcium transient and intracellular sodium dynamics as opposed to transmembrane potential dynamics to constrain the model is important, because when models are fit to replicate physiological phenomena, the final model parameters critically depend on what physiological characteristics are used to constrain the model. To test their model, the authors used experiments using the Hybrid Mouse Diversity Panel.The authors state at the end of their discussion, that this work has very broad implications in the realm of understanding how drugs will affect cardiac health in a diverse population. Since ion channel expression levels can span a wide range (both among individuals in a population as well as between cells in a single individual), the effect of a drug that binds to a specific ion channel could vary widely between individuals or from cell to cell. Therefore, to accurately characterize the effects of a drug that interacts with cardiac ion channels across an entire population, it is important to fully characterize the parameter space of physiologically allowable ionic conductances. This is an important point and could be emphasized more.What seems to be missing from the study at present is a description of the fundamental insight that is gained by the Ca-based GES search compared to the V-based GES search. It's hard to imagine that the V-based search wouldn't reveal the relationship between I_to,f+I_Kur and I_CaL. Although I understand that the Ca signal is (likely) more directly related to ion-channel regulation, there is no particular control mechanism included in the analysis – so what does the current state analysis provide that the V-based doesn't? It would be so useful to show a direct comparison of the Ca- and V-based GES searches. It should also be mentioned that there are a host of regulatory mechanisms not accounted for in either search method.

As noted in the response to reviewer 2, we have added new simulations directly comparing Ca^2+^ sensing alone with voltage sensing alone (Figure 3—figure supplement 3). With voltage sensing alone, the AP waveform was readily constrained as expected, but the CaT was highly variable and often nonphysiological. The Ca/K current ratio relationship was no longer preserved, since other inward and outward currents could regulate AP duration when the CaT was not constrained. As noted above, our hypothesis provides a clear physiological link for Ca^2+^ sensing, since the CaT is critical for generating blood pressure, which is sensed by the carotid baroreceptors and feeds back through the autonomic nervous system to regulate the CaT by controlling levels of Ca-cyling proteins and the AP in a way that preserves blood pressure. This provides a very straightforward physiological mechanism that not only constrains the CaT to a physiological waveform, but also, as an added and novel bonus, constrains AP features through the Ca current-K current ratio. It is much less clear, on the other hand, how voltage would be sensed by the heart to provide a feedback mechanism to control both the AP waveform and the CaT. Voltage-sensing alone provided a reliable AP waveform, but a highly unreliable CaT, as now shown in the new results. We have emphasized these points in the revised text.

1) More discussion of the range of APs waveforms observed would be helpful, since that is what has been examined in all other cardiac AP population approaches. Additionally, in the Discussion section ("physiological AP waveform whose duration matches the experimentally observed variation in isolated myocytes even though the voltage signal is not used to constrain model parameters"), doesn't seem to be entirely supported by the data shown. Figure 3B would better support this argument if the full range of simulated AP behaviors were quantified and compared to experimental AP data. It is not clear how the APs shown in Figure 3B (and later in Figure 5) were chosen, or if they represent the full range of behaviors observed with this approach.

As noted above in the response from reviewer 2, it is clear from visual inspection of the experimental traces in Figure 5—figure supplement 1 that there is marked variation in the AP waveform among different myocytes, ranging from about 10 to 80 ms. This is comparable to the range of APD values in Figure 5E, although there is not sufficient experimental data to construct a histogram for direct comparison. In the text, we have modified the text to state: “A remarkable and nontrivial finding of the present computational study is that Ca^2+^ sensing suffices to produce a physiological AP waveform whose duration spans comparable range (see Figure 5C) to that recorded experimentally in isolated myocytes (Figure 5—figure supplement 1), even though the voltage signal is not used to constrain model parameters.”

2) Similarly, for Figure 5C, it would be clearer if cell-to-cell variability in the simulated APs was quantified and compared that to the experimental range. It seems that the variability in the AP waveforms shown in Figure 5C represent a much wider range of morphologies (in APD_90_, for example), than in the experimental APs shown in Figure 5—figure supplement 1.

See the response to the previous comment.

3) It is not clear what is defined as the "normal CaT (Calcium Transient)", or how that was decided. The reference values (G_ref_) are defined as the conductance in the baseline model, which produce the normal CaT. Is the normal CaT the same as the reference outputs defined in Table 6? Are these the only two outputs used to define the normal CaT? Also, for the reference CaT outputs (as shown in Table 6), how were these values decided? Include a citation (or justification) for the source of the reference/normal values.

We arbitrarily constrained the steady-state CaT amplitude, [Ca]i, time averaged

cytosolic Ca^2+^concentration, and the intracellular sodium concentration [Na]_i_to fall within a 5% tolerance of the control values at a 4 Hz pacing frequency. Slightly increasing or decreasing the tolerance only increases or decreases the range of conductances producing a normal CaT but does not destroy the correlation between Ca and K currents shown in Figures 3F and Figure 4C. Furthermore, as explained in response to a similar question of reviewer #2, we chose a 4 Hz pacing frequency based on the fact that the Bondarenko et al., ventricular myocyte mouse model used in the present study was benchmarked against experimental measurements of CaT amplitude (Figure 18 in Bondarenko et al., 2018) for frequencies ranging from 0.5 to 6 Hz. Since different model parameters corresponding to different strains reproduce similar CaT amplitudefrequency curves as the Bondarenko et al., model, we do not expect the choice of pacing frequency to be critically important for calibrating model parameters that produce a normal electrophysiological phenotype. We have discussed this point in the Discussion section.

4) In the populations shown in Figure 5, is a similar coupling effect seen for average cytosolic Ca^2+^ over one beat? Does this range of simulated values, for either CaT measure, mimic the experimental range?

Yes, a similar coupling effect is seen for the average cytosolic Ca^2+^ concentration and is stronger than for the CaT amplitude. We have added a new Figure (Figure 5—figure supplement 2) to show this effect. We did not measure CaTs experimentally, which are very hard to quantify accurately to compare with the simulated data.

5) My understanding is that the point of Figure 5 is that even if the conductances considered here aren't tightly correlated, the Ca2+ transient and especially the AP are constrained due to electrical coupling in tissue. However, what is not clear to me based on the authors' discussion is if electrical coupling is enough to constrain the Ca2+ transient and AP to within a healthy range. Based on looking at the figure, electrical coupling does not constrain the Ca2+ transient to within the 5% error used in the earlier good enough solutions. But how well constrained do the Ca^2+^ transient and AP really need to be? Also, how would things change if the parameter sets of the cells in the tissue came from the good enough solution?

We apologize for the confusion here. In the GES search, the CaT was constrained to within 5%. However, in the patch clamp experiments, the K and Ca currents had to be measured in different myocytes to isolate the currents properly, so that we could not verify whether the Ca/K current ratio was constrained at the single myocyte level, even though the average ratio for all of the myocytes studied was constant. In this simulation, we addressed the issue of whether a constant Ca/K current ratio at the single myocyte level was important or not when extrapolated to the tissue level. Therefore, we assigned random experimentally measured values of the Ca current and K current densities for the C57BL/6 strain (blue point cluster in Figure 5B) to the AP model to generate the AP and CaT distributions shown in Figure 5E. Because the Ca/K current ratio was not constrained in this population of models, both APD and CaT showed marked variation (unlike the case for the GES search in which the CaT was within 5%). Under these conditions, the APD variance was markedly reduced by coupling, and the CaT variance was modestly reduced. Thus, this tissue simulation represents a worse-case scenario in which the Ca/K current ratio was unconstrained at the single myocyte level, but the mean Ca/K current ratio from many myocytes was constrained. The conclusion is that in coupled tissue, marked variation is the APD can be compensated for at the tissue level by coupling, and marked variation in the CaT can be somewhat compensated for by coupling. However, although we could not ascertain whether the Ca/K current ratio is constant at the single myocyte level to produce low CaT variance (e.g. within 5%), the cell shortening data in Figure 4B shows that in the C57BL/6 strain, cell shortening (as a surrogate for CaT amplitude) varied about 2-fold. This is considerably less than the variance in the CaT amplitude distribution for the uncoupled simulated AP models in Figure 5E, in which the Ca/K current ratio was completely unconstrained. We have revised subsection “Compensation at the organ scale” to clarify these points.

[Editors' note: further revisions were requested prior to acceptance, as described below.]

The manuscript has been improved but there are some remaining issues that we would like you to address before acceptance, as outlined below. Although your revised manuscript will not be sent to the reviewers, we encourage you to carefully consider their suggestions.Reviewer #1:The authors have done a good job of addressing the general concerns that I raised in my original review. Notably, they have sharpened up the presentation of the overall hypothesis of CaT regulation, via blood-pressure feedback, serving as the mechanism governing underlying levels of Ca-cycling proteins and the action potential. As I noted in my original review, the computational findings are supported by analyses of the Hybrid Mouse Diversity Panel, which strengthen the study significantly and set it apart from a purely computational study.One concern I have relates to a point made in response to my review, as well as the other reviewers (and in the Discussion section of the revised manuscript). The authors provide results and analyses to defend the approach of focusing on calcium for the GES determination, rather than also incorporating the action potential. They argue that the variation in the action potential waveform that occurs using the CaT-based parameterization is within expected physiological variability, especially if we expand our definition of physiological variability to include outlier AP waveforms that are excluded via selection bias by experimentalists who "often" discard sick-looking AP waveforms. My first problem here is that this statement is an over simplification of what good experimentalists do and why. Sure, some may just exclude APs that look wrong, but that is not good accepted practice. Instead, they use specific quantifiable criteria, such as minimum resting membrane potential, which have been demonstrated to be correlated to damaged cells, for exclusion. While such criteria may still be excluding physiological APs, my point is that it is more nuanced than the text implies and, I think, does a disservice to the actual experiments and the reported data. Beyond that issue, within the very same paragraph, the authors note that the variability that occurs in CaT using voltage-based parameterization is "highly variable and often nonphysiological." So why is it that the variability that arises using the approach that fits with the authors hypothesis is physiological, whereas the variability using the approach that does not fit the authors hypothesis is "nonphysiological?" Put another way, when the authors define nonphysiological CaT, are they assuming that experimentalists who recorded and reported CaT have never thrown away "sick looking" calcium transient data, which, if included in the literature, would expand the range of physiological? This point is really just about framing, rather than about the overall justification of the approach, as I do agree with the overall conclusion that these findings support using CaT. But I think that point can be made without the "sick looking" discussion or expanding that discussion with more context.

We agree with the reviewer that our main point that the CaT suffices to regulate ion channel conductances without voltage sensing can be made without invoking “sick looking” cells. Accordingly, we have replaced the part of the text in the Discussion section:

“Experimentalists often exclude atypical “sick-looking'' AP waveforms in patch clamp experiments, which are assumed to be “non-physiological'' due to cell damage during enzymatic isolation or patching. Thus, AP waveform variation in isolated myocytes may be underestimated because of this selection bias. In coupled tissue, however, occasional atypical AP waveforms would be voltage-clamped by their “healthy'' AP waveform neighbors to produce an overall uniform AP waveform.”

by

“This finding is consistent with the observations that AP variability is considerable when measured experimentally in patch clamp studies (Figure 5—figure supplement 1), but is greatly reduced in tissue because less frequent atypical AP waveforms are voltage-clamped by the more typical AP waveforms of their neighbors.”

I agree with the point made by referee #2 that it would be useful for the authors to deposit the code used to generate the results.

The codes that produced the results have been uploaded to a public repository cited in the article as

Code availability

The codes used in this work are available at:

https://github.com/circs/GES

Reviewer #2:The authors have followed most of our suggestions and the paper is stronger. The title remains too generic and would benefit from at least mentioning calcium as the main novelty of the paper (and perhaps mice too). Otherwise it could very well be another review on the topic.

Following the suggestion of the reviewer, we have modified the title to refer explicitly to the calcium transient as the main novelty of our study and to specify that it was carried out in mouse populations (i.e. using the Hybrid Mouse Diversity Panel).